# On the Similarities of Embeddings in Contrastive Learning

Chungpa Lee [1]  Sehee Lim [1]  Kibok Lee [1]  Jy-yong Sohn [1]

## Abstract

Contrastive learning operates on a simple yet effective principle: Embeddings of positive pairs are pulled together, while those of negative pairs are pushed apart. In this paper, we propose a unified framework for understanding contrastive learning through the lens of cosine similarity, and present two key theoretical insights derived from this framework. First, in full-batch settings, we show that perfect alignment of positive pairs is *unattainable* when negative-pair similarities fall below a threshold, and this misalignment can be mitigated by incorporating within-view negative pairs into the objective. Second, in mini-batch settings, smaller batch sizes induce stronger separation among negative pairs in the embedding space, i.e., *higher variance* in their similarities, which in turn degrades the quality of learned representations compared to full-batch settings. To address this, we propose an auxiliary loss that reduces the variance of negative-pair similarities in mini-batch settings. Empirical results show that incorporating the proposed loss improves performance in small-batch settings.

## 1. Introduction

Contrastive learning (CL) has emerged as a powerful approach to representation learning (Chen & He, 2021; Chen et al., 2020; He et al., 2020; Radford et al., 2021; Zhai et al., 2023; Khosla et al., 2020). In CL, an embedding model is trained to produce similar representations for two different views of the same data instance—referred to as a positive pair—and dissimilar representations for views from different data instances—referred to as negative pairs. Recent empirical studies have shown that representations learned by these objectives, *i.e., aligning* positive pairs while *separat-*

*ing* negative pairs, achieve remarkable performance across various downstream tasks (Wang & Isola, 2020).

In parallel, several theoretical studies have analyzed the embeddings obtained by CL in full-batch settings. Lu & Steinerberger (2022) showed that the widely used InfoNCE loss (Oord et al., 2018) achieves its minimum value when positive pairs are perfectly aligned, and negative pairs are uniformly separated with the cosine similarity of $-\frac{1}{n-1}$, where $n$ is the size of the training dataset. Lee et al. (2024) further extended this optimality analysis to other contrastive losses, including the SigLIP loss (Zhai et al., 2023).

Due to computational constraints, CL methods are typically implemented using mini-batches rather than full batches in practical scenarios. This has motivated recent studies on understanding whether the property of the optimal state observed in full-batch settings, *i.e.,* perfect alignment of positive pairs and uniform separation of negative pairs with the similarity of $-\frac{1}{n-1}$, also holds under mini-batch settings. Cho et al. (2024) partially addressed this by showing that, for the InfoNCE loss, the optimal embeddings of mini-batch settings are identical to those of full-batch settings, only when the sum of all possible mini-batch losses is minimized. Koromilas et al. (2024) explored embeddings learned through kernel-based contrastive losses (Li et al., 2021; Waida et al., 2023) in mini-batch settings. While these studies provide valuable insights into understanding CL, their analyses are limited to specific forms of contrastive losses.

To address this limitation, we propose a unified framework for analyzing the embeddings obtained by both full-batch and mini-batch settings. Our analysis centers on the cosine similarity of embeddings of both positive and negative pairs. By characterizing the statistical properties of similarities of these embeddings, we reveal how different contrastive losses influence the structure of the learned embedding space under various training conditions.

Our key contributions are summarized as follows:

- In full-batch settings, we identify a fundamental trade-off between the *alignment* of positive pairs and the *separation* of negative pairs. Specifically, we show that the perfect alignment of positive pairs is not feasible when the average similarity of negative pairs falls below the certain threshold $-\frac{1}{n-1}$. We demonstrate that such misalignment

---

[1]Department of Statistics and Data Science, Yonsei University, Seoul, Korea. Correspondence to: Jy-yong Sohn <jysohn1108@yonsei.ac.kr>.

*Proceedings of the 42$^{nd}$ International Conference on Machine Learning*, Vancouver, Canada. PMLR 267, 2025. Copyright 2025 by the author(s).

arises in a class of existing contrastive losses, which can be mitigated by incorporating within-view negative pairs into the contrastive loss.

- In mini-batch settings, we demonstrate that negative pairs within the same batch exhibit stronger separation compared to those from different batches. As a result, we show that smaller batch sizes induce a higher variance in the similarities of negative pairs in the embedding space. We identify this increased variance as a distinctive feature of mini-batch settings that is absent in full-batch settings.

- Motivated by prior studies that full-batch settings often outperform their mini-batch counterparts, we hypothesize that the increased variance may underlie this performance gap. To explore this, we propose an auxiliary loss that can be integrated into contrastive losses to reduce this variance. Empirical results show that incorporating the proposed term improves the performance of CL methods, especially in small-batch settings.

## 2. Related Work

**Contrastive Loss.** The InfoNCE loss (Gutmann & Hyvärinen, 2010; Oord et al., 2018) is a widely adopted contrastive loss that has been applied to various tasks (Wu et al., 2018; Hjelm et al., 2019; Bachman et al., 2019; Chi et al., 2021; Gao et al., 2021; Qian et al., 2021). SimCLR (Chen et al., 2020) modifies the InfoNCE loss to improve robustness to augmentations by treating different augmented views of the same instance as positives and all other augmented instances in the batch as negatives. However, SimCLR simultaneously optimizes both positive and negative pairs in the normalization, which can introduce conflicts during optimization. To address this issue, Decoupled Contrastive Loss (DCL) (Yeh et al., 2022) modifies the normalization so that the selection of negative pairs is restricted. Building on DCL, Decoupled Hyperspherical Energy Loss (DHEL) (Koromilas et al., 2024) further refines the selection by focusing on negative pairs between augmented views of the same instance, which are more challenging than those between different instances. Figure 5 visualizes which pairs are included as positives and negatives for each method.

Meanwhile, Zhai et al. (2023) raised concerns about the softmax function in the InfoNCE loss, noting that it causes all instances to be dependent on each other through normalization. To address this limitation, they propose replacing the softmax with a sigmoid function, which allows each instance to be processed independently in an additive manner.

**Understanding CL Through Embedding Structures.** Several studies have examined how optimal embeddings should be structured to minimize contrastive loss (Lee et al., 2025). Lu & Steinerberger (2022) showed that the optimal

embeddings that minimize the InfoNCE loss form a simplex Equiangular Tight Frame (ETF) (Papyan et al., 2020; Sustik et al., 2007), where each positive pair is perfectly aligned and negative pairs are equally separated at the same angle, resulting in maximal separation among representations. Lee et al. (2024) further showed that the sigmoid-based contrastive loss, a variant of the softmax-based InfoNCE loss, also achieves the same simplex ETF optimum when the temperature parameter of the loss is sufficiently large. This optimal simplex ETF structure still holds even in mini-batch settings, provided that optimization is performed over all possible mini-batch combinations, rather than a single batch at a time (Cho et al., 2024). Building on these findings, we introduce a unified theoretical analysis of the similarities of embedding pairs.

**Effect of Batch Size in CL.** CL shows outstanding performance, particularly when trained with large batch sizes (Chen et al., 2020; Radford et al., 2021; Pham et al., 2023; Tian et al., 2020b; Jia et al., 2021). However, large batch sizes require substantial memory resources, which poses practical challenges and often necessitates the use of smaller batches. This compromise in batch size typically leads to performance degradation, motivating several theoretical studies to investigate its causes (Cho et al., 2024; Koromilas et al., 2024). For example, Yuan et al. (2022) demonstrate that the optimization error in SimCLR (Chen et al., 2020) is upper bounded by a function inversely proportional to the batch size, indicating that smaller batches yield larger optimization errors. Additionally, Chen et al. (2022) show that contrastive losses exhibit increasing discrepancies between the true gradients and those estimated during training as the batch size decreases. While previous studies have primarily focused on optimization error and gradient estimation, we prove that training with small batch sizes leads to increased variance in the similarities of negative pairs in learned embeddings.

## 3. Problem Setup

Let $(\boldsymbol{x}, \boldsymbol{y})$ denote a pair of data points used for model training, where $\boldsymbol{x}$ and $\boldsymbol{y}$ correspond to two distinct views of instances. This formulation provides a unified framework for CL in both unimodal (Chen et al., 2020) and multimodal (Radford et al., 2021) settings. In the unimodal case, $\boldsymbol{x}$ and $\boldsymbol{y}$ are two randomly augmented views. In the multimodal case, $\boldsymbol{x}$ and $\boldsymbol{y}$ are views from different modalities. For clarity, we present our analysis in the unimodal setting, but the findings also apply to the multimodal case.

In CL, an encoder $f(\cdot) \in \mathbb{R}^d$ is trained to map inputs into $d$-dimensional embedding vectors, thereby representing the data. The encoder is assumed to produce normalized embeddings such that $\|\boldsymbol{u}\|_2 = 1$ for all embeddings $\boldsymbol{u}$. This

normalization is commonly adopted in related works for mathematical simplicity (Wang et al., 2017; Wu et al., 2018; Tian et al., 2020a; Wang & Isola, 2020; Zimmermann et al., 2021; Cho et al., 2024; Lee et al., 2024), and is widely used in practice, as experimental results consistently demonstrate its effectiveness in improving performance (Chen et al., 2020; Chen & He, 2021; Xue et al., 2024).

The encoder produces outputs $\boldsymbol{u} = f(\boldsymbol{x})$ and $\boldsymbol{v} = f(\boldsymbol{y})$, which together form an *embedding pair* $(\boldsymbol{u}, \boldsymbol{v})$. When the embedding pair is generated from augmented views of the same instance, it is called a *positive embedding pair* and is encouraged to be similar. In contrast, a *negative embedding pair*, where each embedding comes from different instances, is encouraged to be dissimilar. For simplicity, we refer to positive embedding pairs as *positive pairs* and similarly to *negative pairs*, when there is no risk of confusion.

Formally, $p_{\text{pos}}(\boldsymbol{x}, \boldsymbol{y})$ denotes the distribution of positive pairs, and $p_{\text{neg}}(\boldsymbol{x}, \boldsymbol{y})$ represents the distribution of negative pairs. As in prior work (Wang & Isola, 2020), the marginal distribution of augmented view $\boldsymbol{x}$ is denoted by $p_x$, and similarly use $p_y$ to denote the marginal distribution for $\boldsymbol{y}$. These marginals satisfy the following conditions: $p_x(\boldsymbol{x}) = \int p_{\text{pos}}(\boldsymbol{x}, \boldsymbol{y}) d\boldsymbol{y} = \int p_{\text{neg}}(\boldsymbol{x}, \boldsymbol{y}) d\boldsymbol{y}$ for all $\boldsymbol{x}$, and $p_y(\boldsymbol{y}) = \int p_{\text{pos}}(\boldsymbol{x}, \boldsymbol{y}) d\boldsymbol{x} = \int p_{\text{neg}}(\boldsymbol{x}, \boldsymbol{y}) d\boldsymbol{x}$ for all $\boldsymbol{y}$. Note that the randomness in these distributions arises from the data augmentation process used to generate different views.

Let $n$ be the size of the training dataset. For any positive integers $a$ and $b$ with $a \leq b \leq n$, define the index sets $[a : b] := \{a, a + 1, \cdots, b\}$ and $[a] := [1 : a]$, where index $i \in [n]$ refers to the $i$-th instance in the dataset. For $i \in [n]$, let $\hat{p}_{\text{pos}}^i(\mathbf{x}, \mathbf{y})$ denote the empirical distribution of positive pairs derived from the $i$-th instance. We assume that the supports of $\hat{p}_{\text{pos}}^i$ and $\hat{p}_{\text{pos}}^j$ are disjoint for $i \neq j$, as each instance is distinct. Moreover, we assume that each instance is used equally for training. Therefore, the empirical distribution of all positive pairs, denoted as $\hat{p}_{\text{pos}}(\mathbf{x}, \mathbf{y})$, can be written as $\hat{p}_{\text{pos}}(\mathbf{x}, \mathbf{y}) = \frac{1}{n} \sum_{i \in [n]} \hat{p}_{\text{pos}}^i(\mathbf{x}, \mathbf{y})$. Similarly, the empirical distribution of all negative pairs is denoted by $\hat{p}_{\text{neg}}(\mathbf{x}, \mathbf{y})$.

Following the notations introduced in Koromilas et al. (2024), we denote the element-wise pushforward measures induced by the encoder $f$ as $f_{\sharp}\hat{p}_x$, $f_{\sharp}\hat{p}_y$, $f_{\sharp}\hat{p}_{\text{pos}}$, and $f_{\sharp}\hat{p}_{\text{neg}}$. For example, $f_{\sharp}\hat{p}_{\text{neg}}(\mathbf{u}, \mathbf{v})$ represents the empirical distribution of negative embedding pairs, *i.e.,* the distribution of $(\mathbf{u}, \mathbf{v}) = (f(\boldsymbol{x}), f(\boldsymbol{y}))$ where $(\boldsymbol{x}, \boldsymbol{y})$ comes from $\hat{p}_{\text{neg}}$.

**Contrastive Loss.** For any $a \leq b$ with $a, b \in [n]$, let $(\boldsymbol{U}_{[a:b]}, \boldsymbol{V}_{[a:b]}) := \{(\boldsymbol{u}_i, \boldsymbol{v}_i) : i \in [a : b]\}$ be the set of embedding pairs for instances whose indices range from $a$ to $b$. The notation $(\boldsymbol{U}_{[a:b]}, \boldsymbol{V}_{[a:b]}) \sim f_{\sharp}\hat{p}_{\text{pos}}^{[a:b]}$ indicates that each positive pair $(\boldsymbol{u}_i, \boldsymbol{v}_i)$ is sampled from $f_{\sharp}\hat{p}_{\text{pos}}^i$ for all $i \in [a : b]$. Note that $\boldsymbol{u}_i$ and $\boldsymbol{v}_i$ are random variables, as

they are the encoder outputs of randomly augmented views. For simplicity, the subscript is omitted for the set of all $n$ pairs, *i.e.,* $(\boldsymbol{U}, \boldsymbol{V}) := (\boldsymbol{U}_{[n]}, \boldsymbol{V}_{[n]})$.

Let $f^{\star}$ be the optimal encoder that minimizes the expected contrastive loss, given by

$$f^{\star} := \arg\min_f \mathbb{E}_{(\boldsymbol{U}, \boldsymbol{V}) \sim f_{\sharp}\hat{p}_{\text{pos}}^{[n]}} \left[ \mathcal{L}(\boldsymbol{U}, \boldsymbol{V}) \right], \quad (1)$$

where $\mathcal{L}(\boldsymbol{U}, \boldsymbol{V})$ denotes the contrastive loss for a given sample $(\boldsymbol{U}, \boldsymbol{V})$. In this work, we focus on two specific forms of contrastive losses used in practice:

**Definition 3.1** (InfoNCE-Based Contrastive Loss). For a given index set $\boldsymbol{I} \subseteq [n]$, the contrastive loss $\mathcal{L}_{\text{info-sym}}(\boldsymbol{U}_{\boldsymbol{I}}, \boldsymbol{V}_{\boldsymbol{I}})$ is defined in its symmetric form as

$$\mathcal{L}_{\text{info-sym}}(\boldsymbol{U}_{\boldsymbol{I}}, \boldsymbol{V}_{\boldsymbol{I}}) := \frac{1}{2} \mathcal{L}_{\text{info}}(\boldsymbol{U}_{\boldsymbol{I}}, \boldsymbol{V}_{\boldsymbol{I}}) + \frac{1}{2} \mathcal{L}_{\text{info}}(\boldsymbol{V}_{\boldsymbol{I}}, \boldsymbol{U}_{\boldsymbol{I}}), \quad (2)$$

where the asymmetric component $\mathcal{L}_{\text{info}}(\boldsymbol{U}_{\boldsymbol{I}}, \boldsymbol{V}_{\boldsymbol{I}})$ is

$$\mathcal{L}_{\text{info}}(\boldsymbol{U}_{\boldsymbol{I}}, \boldsymbol{V}_{\boldsymbol{I}}) := \frac{1}{|\boldsymbol{I}|} \sum_{i \in \boldsymbol{I}} \psi \Bigg( c_1 \sum_{j \in \boldsymbol{I} \setminus \{i\}} \phi \left( (\boldsymbol{v}_j - \boldsymbol{v}_i)^{\top} \boldsymbol{u}_i \right) \\ + c_2 \sum_{j \in \boldsymbol{I} \setminus \{i\}} \phi \left( (\boldsymbol{u}_j - \boldsymbol{v}_i)^{\top} \boldsymbol{u}_i \right) \Bigg),$$

for some constants $(c_1, c_2) \in \{(0, 1), (1, 0), (1, 1)\}$ and some convex and increasing functions $\phi, \psi : \mathbb{R} \to \mathbb{R}$.

**Definition 3.2** (Independently Additive Contrastive Loss). For a given index set $\boldsymbol{I} \subseteq [n]$, the contrastive loss $\mathcal{L}_{\text{ind-add}}(\boldsymbol{U}_{\boldsymbol{I}}, \boldsymbol{V}_{\boldsymbol{I}})$ is defined as

$$\mathcal{L}_{\text{ind-add}}(\boldsymbol{U}_{\boldsymbol{I}}, \boldsymbol{V}_{\boldsymbol{I}}) := -\frac{1}{|\boldsymbol{I}|} \sum_{i \in \boldsymbol{I}} \phi(\boldsymbol{u}_i^{\top} \boldsymbol{v}_i) \quad (3)$$
$$+ \frac{c_1}{|\boldsymbol{I}|(|\boldsymbol{I}| - 1)} \sum_{i \neq j \in \boldsymbol{I}} \psi(\boldsymbol{u}_i^{\top} \boldsymbol{v}_j)$$
$$+ \frac{c_2}{2|\boldsymbol{I}|(|\boldsymbol{I}| - 1)} \sum_{i \neq j \in \boldsymbol{I}} \left( \psi(\boldsymbol{u}_i^{\top} \boldsymbol{u}_j) + \psi(\boldsymbol{v}_i^{\top} \boldsymbol{v}_j) \right)$$

for some constants $(c_1, c_2) \in \{(0, 1), (1, 0), (1, 1)\}$, where $\phi : \mathbb{R} \to \mathbb{R}$ is a differentiable, concave, and increasing function, and $\psi : \mathbb{R} \to \mathbb{R}$ is a differentiable, convex, and increasing function. Here, $i \neq j \in \boldsymbol{I}$ is a simplified notation representing $i \in \boldsymbol{I}$ and $j \in \boldsymbol{I} \setminus \{i\}$.

The constants $(c_1, c_2) \in \{(0, 1), (1, 0), (1, 1)\}$ in both loss formulations determine which types of negative pairs are included in the contrastive loss. The *cross-view negatives* refer to pairs of embeddings from different views of different instances, *i.e.,* $(\boldsymbol{u}_i, \boldsymbol{v}_j)$ with $i \neq j$, whereas *within-view negatives* are pairs from the same view but different instances, *i.e.,* $(\boldsymbol{u}_i, \boldsymbol{u}_j)$ or $(\boldsymbol{v}_i, \boldsymbol{v}_j)$ with $i \neq j$ (Shen et al., 2016). Setting $c_1 = 1$ includes cross-view negatives, while $c_2 = 1$

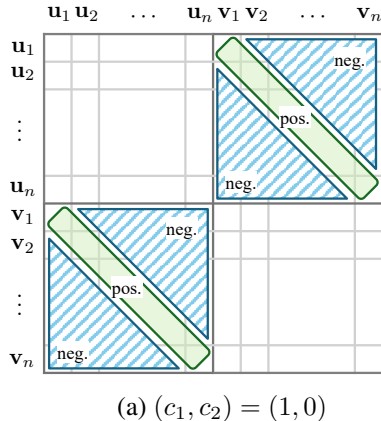 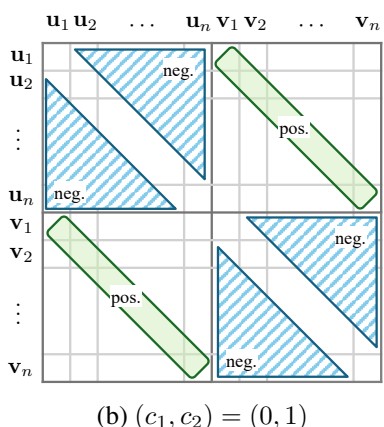 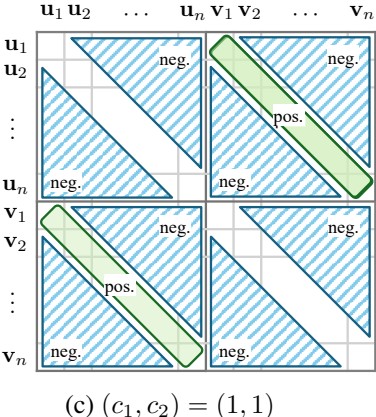

(a) $(c_1, c_2) = (1, 0)$  (b) $(c_1, c_2) = (0, 1)$  (c) $(c_1, c_2) = (1, 1)$

*Figure 1.* Illustration of negative pair considered in the loss formulations defined in Def. 3.1 and Def. 3.2, which depends on the choice of $(c_1, c_2)$. Each grid shows all possible pairs of embeddings in $\boldsymbol{U}_{[n]}$ and $\boldsymbol{V}_{[n]}$, and each cell represents one pair. Green regions represent positive pairs, and blue-striped regions indicate which negative pairs are included in the loss.

includes within-view negatives. Figure 1 summarizes which types of negatives are incorporated for each configuration of $(c_1, c_2)$. Further discussion on the distinction between cross-view and within-view negatives in the single-modal case are provided in Appendix B, emphasizing that their key difference lies in how they are structurally incorporated into the loss, not in how they are generated.

**Remark 3.3.** The first form of losses in Def. 3.1 encompasses a variety of contrastive losses such as InfoNCE (Oord et al., 2018; Radford et al., 2021), SimCLR (Chen et al., 2020), DCL (Yeh et al., 2022), and DHEL (Koromilas et al., 2024), see Appendix A.1. The second form in Def. 3.2 includes contrastive losses such as SigLIP (Zhai et al., 2023) and Spectral CL (HaoChen et al., 2021), see Appendix A.2.

The difference between the two forms of losses lies in computational efficiency. The first form in Def. 3.1 necessitates simultaneous computation based on pairwise similarities across the entire set of embeddings due to the need for normalization, which becomes impractical for extremely large batch sizes. In contrast, the second form in Def. 3.2 is independently additive, allowing it to compute components of each pairwise similarity individually and aggregate them, making it applicable to larger datasets.

## 4. Similarities of Embedding Pairs

In CL, the encoder is trained to bring positive embedding pairs closer together while pushing negative embedding pairs further apart. Accordingly, the cosine similarities of embedding pairs provide a straightforward way to evaluate how well representation achieves its objective. These similarities are formally defined as follows.

**Definition 4.1** (Similarities of Positive/Negative Pairs)**.** The

similarity of embeddings of a positive pair is defined as

$$\boldsymbol{s}(f; \hat{p}_{\text{pos}}) := f(\boldsymbol{x})^\top f(\boldsymbol{y}) \qquad \text{for} \quad (\boldsymbol{x}, \boldsymbol{y}) \sim \hat{p}_{\text{pos}},$$

dubbed as the *positive-pair similarity* for the encoder $f$. Similarly, the similarity of embeddings of a negative pair is defined as

$$\boldsymbol{s}(f; \hat{p}_{\text{neg}}) := f(\boldsymbol{x})^\top f(\boldsymbol{y}) \qquad \text{for} \quad (\boldsymbol{x}, \boldsymbol{y}) \sim \hat{p}_{\text{neg}},$$

dubbed as the *negative-pair similarity* for the encoder $f$. We call both similarities as *embedding similarities*.

Note that the similarities of embeddings in Def. 4.1, denoted by $\boldsymbol{s}(f; \hat{p}_{\text{pos}})$ and $\boldsymbol{s}(f; \hat{p}_{\text{neg}})$, are random variables, where randomness arises from data sampling and augmentation. Specifically, a positive pair is generated by selecting an instance from a dataset of size $n$ and applying two random augmentations. Similarly, a negative pair is generated by randomly selecting an instance pair from the $n(n-1)$ possible combinations and independently applying random augmentations to each instance.

**Expectation & Variance of Negative-pair Similarities.** Recall that the negative-pair similarity $\boldsymbol{s}(f; \hat{p}_{\text{neg}})$ is a random variable. Here we investigate how the expectation and the variance of $\boldsymbol{s}(f; \hat{p}_{\text{neg}})$ affect the learned embeddings. First, as the expectation $\mathbb{E}\left[\boldsymbol{s}(f; \hat{p}_{\text{neg}})\right]$ increases, the negative pairs are less separated, which typically degrades the representation quality. Second, the high variance $\text{Var}\left[\boldsymbol{s}(f; \hat{p}_{\text{neg}})\right]$ implies that some negative pairs are mapped unusually close, while others are mapped much farther apart. Figure 2 shows the effect of the variance $\text{Var}\left[\boldsymbol{s}(f; \hat{p}_{\text{neg}})\right]$ of the negative-pair similarities. Here, we have three different cases of learned embeddings $\{\boldsymbol{u}_i, \boldsymbol{v}_i\}_{i \in [4]}$ in three dimensional space. While all three cases have same expectation $\mathbb{E}\left[\boldsymbol{s}(f; \hat{p}_{\text{neg}})\right]$, the geometry

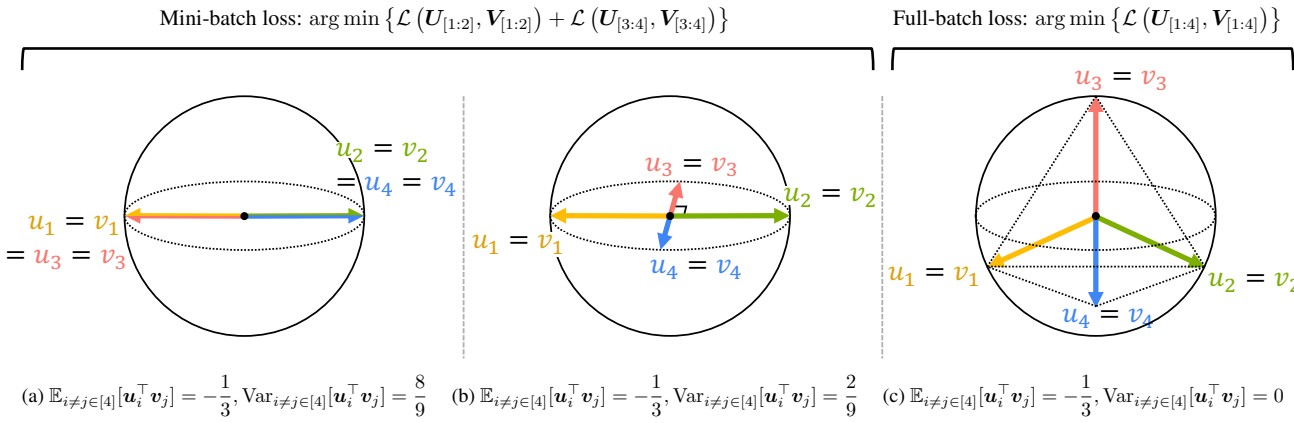

Mini-batch loss: $\arg\min\left\{\mathcal{L}\left(\boldsymbol{U}_{[1:2]},\boldsymbol{V}_{[1:2]}\right)+\mathcal{L}\left(\boldsymbol{U}_{[3:4]},\boldsymbol{V}_{[3:4]}\right)\right\}$

Full-batch loss: $\arg\min\left\{\mathcal{L}\left(\boldsymbol{U}_{[1:4]},\boldsymbol{V}_{[1:4]}\right)\right\}$

(a) $\mathbb{E}_{i\neq j\in[4]}[\boldsymbol{u}_i^\top\boldsymbol{v}_j]=-\frac{1}{3},\mathrm{Var}_{i\neq j\in[4]}[\boldsymbol{u}_i^\top\boldsymbol{v}_j]=\frac{8}{9}$  (b) $\mathbb{E}_{i\neq j\in[4]}[\boldsymbol{u}_i^\top\boldsymbol{v}_j]=-\frac{1}{3},\mathrm{Var}_{i\neq j\in[4]}[\boldsymbol{u}_i^\top\boldsymbol{v}_j]=\frac{2}{9}$  (c) $\mathbb{E}_{i\neq j\in[4]}[\boldsymbol{u}_i^\top\boldsymbol{v}_j]=-\frac{1}{3},\mathrm{Var}_{i\neq j\in[4]}[\boldsymbol{u}_i^\top\boldsymbol{v}_j]=0$

*Figure 2.* Visualization of three different cases of eight embeddings on the three-dimensional unit sphere. Positive embedding pairs are represented in the same color and share the same subscript, while negative pairs refer to any two embeddings with different subscripts. In (a) and (b), embeddings minimize the fixed mini-batch contrastive loss described in Theorem 5.5, with the batches partitioned as $\{\boldsymbol{u}_1,\boldsymbol{v}_1,\boldsymbol{u}_2,\boldsymbol{v}_2\}$ and $\{\boldsymbol{u}_3,\boldsymbol{v}_3,\boldsymbol{u}_4,\boldsymbol{v}_4\}$. In (c), the embeddings minimize the full-batch contrastive loss described in Theorem 5.1. In all cases, the expectation of negative-pair similarities, $\mathbb{E}_{i\neq j\in[4]}[\boldsymbol{u}_i^\top\boldsymbol{v}_j]$, remains the same. However, the variance of negative-pair similarities, $\mathrm{Var}_{i\neq j\in[4]}[\boldsymbol{u}_i^\top\boldsymbol{v}_j]$, increases in mini-batch settings, indicating that some negative pairs are much more similar to each other while others are more dissimilar.

of embeddings significantly changes depending on the variance $\mathrm{Var}\left[\boldsymbol{s}(f;\hat{p}_{\mathrm{neg}})\right]$. One can confirm that the rightmost case having equi-distant embeddings $\{\boldsymbol{u}_i\}_{i\in[4]}$ achieves the *zero* variance of negative-pair similarities.

**Comparison with Existing Metrics.**   The similarities of embeddings in Def. 4.1 are related with metrics proposed in previous work. For example, the *alignment* metric used in (Wang & Isola, 2020) can be represented as

$$\mathbb{E}_{(\boldsymbol{u},\boldsymbol{v})\sim f_\sharp\hat{p}_{\mathrm{pos}}}\left[\|\boldsymbol{u}-\boldsymbol{v}\|_2^2\right]=-2\mathbb{E}\left[\boldsymbol{s}(f;\hat{p}_{\mathrm{pos}})\right]+2,\quad(4)$$

which is related with the positive-pair similarity $\boldsymbol{s}(f;\hat{p}_{\mathrm{pos}})$. Here, the higher *alignment* value indicates that representations are largely invariant to random noise factors introduced by augmentations.

On the other hand, the *uniformity* metric used in (Wang & Isola, 2020) is related with the negative-pair similarity $\boldsymbol{s}(f;\hat{p}_{\mathrm{neg}})$, since it is defined by the logarithm of the Gaussian potential function (Cohn & Kumar, 2007) as

$$\log\mathbb{E}_{\substack{\boldsymbol{u}\sim f_\sharp\hat{p}_x\\\boldsymbol{v}\sim f_\sharp\hat{p}_y}}\left[\exp\left(-\|\boldsymbol{u}-\boldsymbol{v}\|_2^2\right)\right]\quad(5)$$

$$\approx 2\left(\mathbb{E}\left[\boldsymbol{s}(f;\hat{p}_{\mathrm{neg}})\right]+\mathrm{Var}\left[\boldsymbol{s}(f;\hat{p}_{\mathrm{neg}})\right]-1\right),\quad(6)$$

where the approximation in (6) is detailed in Appendix C.1. The *uniformity* metric measures how representations are uniformly distributed on the unit hypersphere, and thus a lower uniformity value indicates that the representations preserve more information from the data.

## 5. Behavior of Learned Embeddings

In this section, we interpret the behavior of embeddings trained by contrastive learning, through the lens of similarities of positive/negative pairs, denoted by $\boldsymbol{s}(f;\hat{p}_{\mathrm{pos}})$ and $\boldsymbol{s}(f;\hat{p}_{\mathrm{neg}})$, defined in Sec. 4. We begin by examining the full-batch CL and subsequently extend our analysis to the mini-batch CL. Proofs for all statements in Sec. 5.1 and Sec. 5.2 are provided in Appendix C.3 and C.4, respectively.

### 5.1. Full-Batch Contrastive Learning

Recall that we consider various contrastive losses which can be categorized into two parts: the InfoNCE-based contrastive loss in Def. 3.1 and the independently additive contrastive loss in Def. 3.2. Our first main result below provides the similarities of positive/negative pairs for the two types of contrastive losses, when the encoder is full-batch trained.

**Theorem 5.1.** *Suppose $d\geq n-1$. Let the contrastive loss $\mathcal{L}(\boldsymbol{U},\boldsymbol{V})$ be one of the following forms:*

*i. $\mathcal{L}_{\mathrm{info\text{-}sym}}(\boldsymbol{U},\boldsymbol{V})$ in Def. 3.1.*

*ii. $\mathcal{L}_{\mathrm{ind\text{-}add}}(\boldsymbol{U},\boldsymbol{V})$ in Def. 3.2, where $(c_1,c_2)\in\{(0,1),(1,1)\}$.*

*iii. $\mathcal{L}_{\mathrm{ind\text{-}add}}(\boldsymbol{U},\boldsymbol{V})$ in Def. 3.2, where $(c_1,c_2)=(1,0)$ and $\phi'(1)>\frac{n-2}{2(n-1)}\cdot\psi'\left(-\frac{1}{n-1}\right)$.*

*Then, in full-batch settings, the embedding similarities for the optimal encoder $f^\star$ in (1) are*

$$\boldsymbol{s}(f^\star;\hat{p}_{\mathrm{pos}})=1,\qquad\boldsymbol{s}(f^\star;\hat{p}_{\mathrm{neg}})=-\frac{1}{n-1}.$$

According to Theorem 5.1, all positive pairs achieve the perfect alignment, while all negative pairs are uniformly separated with the cosine similarity of $-\frac{1}{n-1}$. This is a generalized version of previous studies (Lu & Steinerberger, 2022; Cho et al., 2024; Lee et al., 2024; Koromilas et al., 2024), where we extend in two directions. First, the CL loss formulations (Def. 3.1 and Def. 3.2) we considered are a much broader class of losses compared with existing work. Second, we consider the randomness of $n$ embedding pairs in the optimization, where this randomness is introduced through augmentations. Now we analyze the behavior of embeddings for *arbitrary* encoder $f$, that is not necessarily in the optimal status $f^\star$. The below result provides the relationship between the positive-pair similarity and the negative-pair similarity.

**Theorem 5.2.** *For any normalized encoder $f$,*

$$\mathbb{E}\left[\boldsymbol{s}\left(f;\hat{p}_{\text{pos}}\right)\right] \leq 1 + \left(\mathbb{E}\left[\boldsymbol{s}\left(f;\hat{p}_{\text{neg}}\right)\right] + \frac{1}{n-1}\right), \quad (7)$$

*where the equality in (7) holds if and only if* $\text{tr}\left(\text{Var}_{(\boldsymbol{u},\boldsymbol{v})\sim f_\sharp\hat{p}_{\text{pos}}}[\boldsymbol{u}-\boldsymbol{v}]\right) = 0$ *and* $\mathbb{E}_{\substack{\boldsymbol{u}\sim f_\sharp\hat{p}_x \\ \boldsymbol{v}\sim f_\sharp\hat{p}_y}}[\boldsymbol{u}+\boldsymbol{v}] = \boldsymbol{0}$.

Theorem 5.2 highlights the relationship between the expectation of positive-pair similarities, $\mathbb{E}\left[\boldsymbol{s}\left(f;\hat{p}_{\text{pos}}\right)\right]$, and that of negative-pair similarities, $\mathbb{E}\left[\boldsymbol{s}\left(f;\hat{p}_{\text{neg}}\right)\right]$. When the average of negative-pair similarities drops below $-\frac{1}{n-1}$, the positive pairs cannot be fully aligned, which is not desired. We call such phenomenon as the *excessive separation of negative pairs in full-batch CL*, since the average of negative-pair similarities drops below $-\frac{1}{n-1}$ when negative pairs are more separated compared with the optimal status specified in Theorem 5.1. This issue may arise when certain losses are used, as shown in the following theorem.

**Theorem 5.3** (Excessive Separation in Full-Batch CL)**.** *Suppose that $d \geq n$. Consider the contrastive loss $\mathcal{L}(\boldsymbol{U},\boldsymbol{V})$ in the form of $\mathcal{L}_{\text{ind-add}}(\boldsymbol{U},\boldsymbol{V})$ in Def. 3.2, where $(c_1,c_2) = (1,0)$ and*

$$\phi'(1) < \frac{n-2}{2(n-1)} \cdot \psi'\left(-\frac{1}{n-1}\right). \quad (8)$$

*Then, under full-batch settings, the embedding similarities for the optimal encoder $f^\star$ in (1) satisfy*

$$\boldsymbol{s}(f^\star;\hat{p}_{\text{pos}}) < 1, \qquad \boldsymbol{s}(f^\star;\hat{p}_{\text{neg}}) < -\frac{1}{n-1}.$$

The inequality condition on $\phi$ and $\psi$ in (8) explains why the loss $\mathcal{L}_{\text{ind-add}}(\boldsymbol{U},\boldsymbol{V})$ in Def. 3.2 causes the excessive separation of negative pairs. Specifically, $\mathcal{L}_{\text{ind-add}}(\boldsymbol{U},\boldsymbol{V})$ is formulated as the sum of $\psi(\cdot)$ over negative pairs and $-\phi(\cdot)$ over positive pairs. Therefore, $\psi'(-\frac{1}{n-1})$ indicates how much the loss decreases if the negative-pair similarity falls below $-\frac{1}{n-1}$, while $\phi'(1)$ measures how much the loss

increases if the positive-pair similarity drops below 1. When the condition in (8) is satisfied, ignoring the scaling factor, the loss reduction from separating negative pairs beyond the similarity threshold of $-\frac{1}{n-1}$ can be greater than the loss increase from reducing positive-pair similarity below 1. As a result, the optimization process favors pushing negative pairs even further apart, leading to the excessive separation. We provide a specific loss where this issue arises:

**Example 5.4.** *Consider the sigmoid contrastive loss $\mathcal{L}_{\text{sig}}(\boldsymbol{U},\boldsymbol{V})$ (Zhai et al., 2023), defined as*

$$\mathcal{L}_{\text{sig}}(\boldsymbol{U},\boldsymbol{V}) := \frac{1}{n}\sum_{i\in[n]}\log\left(1+\exp\left(-t\boldsymbol{u}_i^\top\boldsymbol{v}_i\right)\cdot\exp(b)\right)$$
$$+\frac{1}{n}\sum_{i\neq j\in[n]}\log\left(1+\exp\left(t\boldsymbol{u}_i^\top\boldsymbol{v}_j\right)\cdot\exp(-b)\right), \quad (9)$$

*where $t > 0$ and $b \in \mathbb{R}$ are hyperparameters. This loss follows the form of $\mathcal{L}_{\text{ind-add}}(\boldsymbol{U},\boldsymbol{V})$ in Def. 3.2, where $(c_1,c_2) = (1,0)$, $\phi(x) = -\log(1+\exp(-tx+b))$, and $\psi(x) = (n-1)\cdot\log(1+\exp(tx-b))$. If hyperparameters $t$ and $b$ are chosen such that*

$$\frac{1+\exp\left(\frac{t}{n-1}+b\right)}{1+\exp(t-b)} < \frac{n-2}{2}, \quad (10)$$

*then embedding similarities for the full-batch optimal encoder $f^\star$ in (1) satisfy*

$$\boldsymbol{s}(f^\star;\hat{p}_{\text{pos}}) < 1, \qquad \boldsymbol{s}(f^\star;\hat{p}_{\text{neg}}) < -\frac{1}{n-1}.$$

Note that (10) is just a rephrase of (8) by plugging in $\phi$ and $\psi$ for the sigmoid contrastive loss. When the hyperparameter $b$ of the sigmoid contrastive loss is sufficiently small, the condition in (10) is satisfied, thus the learned embedding suffers from the excessive separation issue. This can be also explained by the sigmoid contrastive loss formula. In (9), the relative weight of second term (compared to the first term) increases as $b$ decreases. In such case, minimizing the negative-pair similarity becomes more important. Consequently, decreasing $b$ induces the excessive separation of negative pairs.

**Mitigating Excessive Separation.** A natural question that arises is whether the excessive separation of negative pairs in full-batch CL can be mitigated. According to Theorem 5.1 and Theorem 5.3, this issue depends on the specific form of the contrastive loss. In particular, under the independently additive loss $\mathcal{L}_{\text{ind-add}}(\boldsymbol{U},\boldsymbol{V})$, the optimal embeddings do not suffer from excessive separation when $c_2 = 1$ (i.e., case (ii) of Theorem 5.1), whereas the issue arises when $c_2 = 0$ and condition (8) holds.

This observation suggests two potential solutions to the excessive separation problem. The first is to set $c_2 = 1$ in the loss, effectively incorporating within-view negative pairs. The second is to tune hyperparameters such that condition (8) does not hold, *i.e.,* case (iii) in Theorem 5.1. While both approaches are theoretically valid, the former offers a more principled and practical remedy. In contrast, the latter lacks clear guidance for selecting suitable hyperparameters and may incur significant computational overhead. Therefore, we advocate including within-view negative pairs as a straightforward and effective strategy to avoid excessive separation in full-batch CL.

## 5.2. Mini-Batch Contrastive Learning

In Sec. 5.1, we show that when a proper contrastive loss is chosen for full-batch settings, the learned embedding satisfies the following behavior: all negative pairs have the cosine similarity of $-\frac{1}{n-1}$ and all positive pairs are perfectly aligned. What about the practical scenarios when we use mini-batches for training? Suppose $n$ training samples are partitioned into $b$ mini-batches where each batch contains $m := n/b$ samples. Consider training the embeddings by under the *fixed* mini-batch configuration, where $k$-th mini-batch contains the samples with indices in $\boldsymbol{I}_k = \{m(k-1)+1, m(k-1)+2, \cdots, mk\}$. Under this scenario, the below theorem analyzes the behavior of embeddings learned by mini-batch training.

**Theorem 5.5** (Excessive Separation in Mini-Batch CL). *Suppose $d \geq m - 1$. Let the contrastive loss $\mathcal{L}(\boldsymbol{U}, \boldsymbol{V})$ be one of the forms in Theorem 5.1. Define $f^\star_{\text{batch}}$ as the optimal encoder that minimizes the fixed mini-batch loss, given by*

$$f^\star_{\text{batch}} := \arg\min_f \mathbb{E}_{(\boldsymbol{U}, \boldsymbol{V}) \sim f_\sharp \hat{p}_{\text{pos}}^{[n]}} \left[ \sum_{k \in [b]} \mathcal{L}\left(\boldsymbol{U}_{\boldsymbol{I}_k}, \boldsymbol{V}_{\boldsymbol{I}_k}\right) \right],$$

*where $\boldsymbol{I}_k := [m(k-1)+1 : mk]$ for $k \in [b]$. Then, embedding similarities for the optimal encoder $f^\star_{\text{batch}}$ satisfy*

$$\boldsymbol{s}(f^\star_{\text{batch}}; \hat{p}_{\text{pos}}) = 1,$$

$$\mathbb{E}\left[\boldsymbol{s}(f^\star_{\text{batch}}; \hat{p}_{\text{neg}})\right] = -\frac{1}{n-1},$$

$$\text{Var}\left[\boldsymbol{s}(f^\star_{\text{batch}}; \hat{p}_{\text{neg}})\right] \in \left[\frac{n-m}{(m-1)(n-1)^2}, \frac{n(n-m)}{(m-1)(n-1)^2}\right].$$

(11)

*A necessary condition for attaining the minimum variance of negative-pair similarities in (11) is $d \geq b(m-1)$.*

According to Theorem 5.5, the embeddings learned by mini-batch settings have the following behaviors. First, the positive-pair similarity is equal to 1, *i.e.,* all positive pairs are fully aligned. Second, the *expectation* of negative-pair similarities is equal to $-\frac{1}{n-1}$, which happens for full-batch settings as well. Third, unlike full-batch settings, the

negative-pair similarity is *not uniform* across the pairs, *i.e.,* the variance is positive when the mini-batch size $m$ is strictly less than the sample size $n$. Thus, the effect of using mini-batch (compared with using full-batch) is in the increased variance of negative-pair similarities. Throughout the paper, we call such phenomenon as the *excessive separation of negative pairs in mini-batch CL*.

Figure 2 visualizes the effect of using mini-batches, compared with full-batch settings, when $n = 4$ and $m = 2$. For full-batch settings, shown in (c) of Figure 2, the variance of negative-pair similarities is zero, indicating that all negative pairs are equi-distant. In contrast, (a) and (b) of Figure 2 illustrate the embeddings learned by mini-batch settings, where the fixed mini-batches are specified as $\left(\boldsymbol{U}_{[1:2]}, \boldsymbol{V}_{[1:2]}\right)$ and $\left(\boldsymbol{U}_{[3:4]}, \boldsymbol{V}_{[3:4]}\right)$. For both (a) and (b), the variance of negative-pair similarities is positive, where (a) represents the case that achieves the highest variance in (11), and (b) corresponds to the case that achieves the lowest variance.

**Effect of Batch Size.** Note that the variance of negative-pair similarities in (11) depends on the batch size $m$. For example, in the full-batch case where $m = n$, the upper bound in (11) is zero, which is consistent with Theorem 5.1. One can confirm that the upper and lower bounds on the variance is a monotonically decreasing function of $m$, which implies that smaller batch sizes inherently exacerbates the excessive separation of negative pairs in mini-batch settings.

The below theorem analyzes the effect of batch size on the training dynamics, when a popular CL loss is used:

**Theorem 5.6.** *Consider the InfoNCE loss $\mathcal{L}_{\text{InfoNCE}}(\boldsymbol{U}, \boldsymbol{V})$ (Oord et al., 2018), which corresponds to the loss $\mathcal{L}_{\text{info-sym}}(\boldsymbol{U}, \boldsymbol{V})$ in Def. 3.1 where $\phi(x) = \exp(x/t)$ for some $t > 0$, $\psi(x) = \log(1+x)$, and $(c_1, c_2) = (1, 0)$. For any two integers $m_1, m_2 \in [n]$ satisfying $m_1 \leq m_2$, the gradient of the InfoNCE loss with respect to the negative-pair similarity satisfies*

$$0 \leq \frac{\partial}{\partial \left(\boldsymbol{u}_i^\top \boldsymbol{v}_j\right)} \mathcal{L}_{\text{InfoNCE}}\left(\boldsymbol{U}_{[m_2]}, \boldsymbol{V}_{[m_2]}\right)$$

$$\leq \frac{\partial}{\partial \left(\boldsymbol{u}_i^\top \boldsymbol{v}_j\right)} \mathcal{L}_{\text{InfoNCE}}\left(\boldsymbol{U}_{[m_1]}, \boldsymbol{V}_{[m_1]}\right) \quad (12)$$

*for any distinct indices $i \neq j \in [m]$. Moreover, the equality in (12) holds if and only if $m_1 = m_2$.*

According to Theorem 5.6, the gradient of the loss with respect to the negative-pair similarity is always non-negative, implying that gradient descent decreases the similarities of negative pairs, thereby pushing them further apart. Notably, the magnitude of this gradient increases as the batch size gets smaller. As a result, negative pairs within each batch exhibit greater separation the batch size gets smaller.

**Proposed Variance Reduction Method.** Prior empirical studies on CL have shown that mini-batch settings often underperform compared to full-batch settings (Chen et al., 2020; Radford et al., 2021). This naturally raises a question: *What is the main factor contributing to this performance degradation in mini-batch settings, and how can it be addressed?* According to Theorem 5.5, from the perspective of cosine similarity of embeddings, the key difference introduced by mini-batch settings lies in the increased variance of negative-pair similarities. Motivated by this theoretical insight, we propose an approach to improve mini-batch contrastive learning by introducing an auxiliary loss term, $\mathcal{L}_{\mathrm{VRNS}}(\mathbf{U}, \mathbf{V})$, which explicitly reduces the variance of negative-pair similarities:

**Definition 5.7** (Reducing Variance of Negative-Pair Similarities). Let $m$ be the mini-batch size. Define

$$\mathcal{L}_{\mathrm{VRNS}}(\mathbf{U}_{[m]}, \mathbf{V}_{[m]}) := \frac{1}{m(m-1)} \sum_{i \neq j \in [m]} \left( \mathbf{u}_i^\top \mathbf{v}_j + \frac{1}{n-1} \right)^2$$

as the auxiliary loss for reducing the variance of negative-pair similarities.

One can combine arbitrary conventional mini-batch loss $\mathcal{L}\left(\boldsymbol{U}_{[m]}, \boldsymbol{V}_{[m]}\right)$ with the proposed auxiliary loss to get the modified loss, given by

$$\mathcal{L}\left(\boldsymbol{U}_{[m]}, \boldsymbol{V}_{[m]}\right) + \lambda \cdot \mathcal{L}_{\mathrm{VRNS}}\left(\boldsymbol{U}_{[m]}, \boldsymbol{V}_{[m]}\right),$$

where $\lambda > 0$ is a hyperparameter. By including the proposed term into the mini-batch loss, we encourage all negative-pair similarities to be close to $-\frac{1}{n-1}$, the ideal value achieved in full-batch settings in Theorem 5.1. As a result, the proposed loss controls the variance of negative-pair similarities.

## 6. Empirical Validation

In this section, we empirically validate the impact of our theoretical results discussed in Sec. 5, especially for the practical scenarios of mini-batch settings. First, we empirically observe that the excessive separation of negative pairs (proven in Theorem 5.5) actually happens in experiments on benchmark datasets. Second, we empirically confirm that such excessive separation issue can be mitigated by using the proposed loss in Def. 5.7 which reduces the variance of the negative-pair similarities. Third, we observe such variance reduction improves the quality of learned representations in various real-world experiments.

### 6.1. Excessive Separation of Negative Pairs

To investigate the excessive separation of negative pairs in CL, we evaluate the variance of the negative-pair similarities of embeddings learned by real-world experiments. Following prior works on contrastive learning (Chen et al., 2020;

*Table 1.* Variance of the similarities of embeddings of negative pairs, obtained from models trained with different batch sizes. Each model is trained using either the SimCLR loss alone or jointly with our auxiliary loss in Def. 5.7, which is proposed to reduce this variance. One can confirm that the variance is effectively reduced by using the proposed auxiliary loss.

| Batch size | Variance of negative-pair similarities | |
| --- | --- | --- |
| | SimCLR | SimCLR + Ours |
| 32 | 0.1649 | 0.1008 |
| 64 | 0.1505 | 0.0952 |
| 128 | 0.1444 | 0.0929 |
| 256 | 0.1404 | 0.0921 |
| 512 | 0.1396 | 0.0917 |

Koromilas et al., 2024), we use a ResNet-18 encoder (He et al., 2016) followed by a two-layer projection head. The models are pretrained on CIFAR-100 (Krizhevsky et al., 2009) by minimizing the SimCLR loss with the temperature parameter of $t = 0.2$. Five models are trained with mini-batches sampled uniformly at random, with batch sizes of 32, 64, 128, 256, and 512, respectively. Additional details of the experimental setup are provided in Appendix D.

Based on these pretrained models, we generate 5,000 positive embedding pairs by applying random augmentations to the training data and extracting the corresponding outputs from the projection head. We then compute the cosine similarities of negative pairs, and report the variance of these similarities in Table 1. As shown in the table, training with smaller batch sizes leads to higher variance in negative-pair similarities, which aligns with the result in Theorem 5.5.

To evaluate the effectiveness of our proposed auxiliary loss, we train five models for each batch size by minimizing the SimCLR loss combined with the auxiliary loss $\mathcal{L}_{\mathrm{VRNS}}(\mathbf{U}, \mathbf{V})$ in Def. 5.7 with the hyperparameter of $\lambda = 30$. The variances of negative-pair similarities from these additional models are shown in the last column of Table 1, and are consistently reduced across all batch sizes. This indicates that our proposed loss effectively mitigates excessive separation of negative pairs in mini-batch settings.

### 6.2. Effect of Variance Reduction on Performance

We further investigate whether reducing the variance of negative-pair similarities improves the quality of learned representations in terms of the downstream performances.

**Experimental Setup.** We pretrain models on CIFAR-10, CIFAR-100 (Krizhevsky et al., 2009), and ImageNet (Deng et al., 2009) using various contrastive losses that follow the formulation in Def. 3.1, including SimCLR, DCL, and DHEL. For all methods, we compare models trained with and without incorporating the auxiliary loss $\mathcal{L}_{\mathrm{VRNS}}(\mathbf{U}, \mathbf{V})$

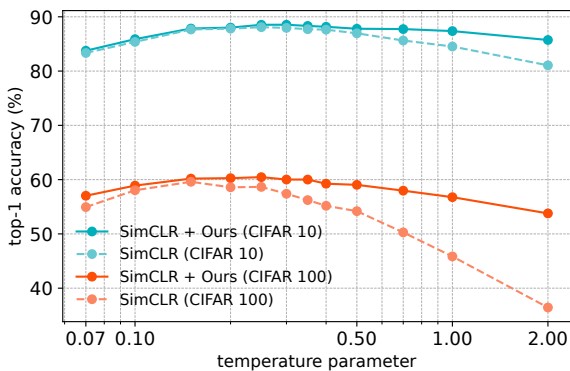

*Figure 3.* Classification accuracy on CIFAR datasets. Models are trained by minimizing the SimCLR loss with and without the auxiliary loss proposed in Def. 5.7, using various temperature parameters in the SimCLR loss.

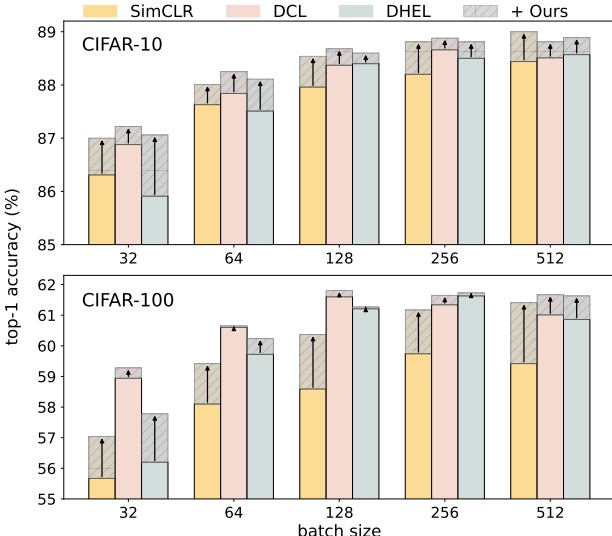

*Figure 4.* Effect of the auxiliary loss proposed in Def. 5.7 on top-1 classification accuracy when combined with various baseline methods (SimCLR, DCL, and DHEL) on CIFAR datasets. The gray bars highlight the performance gains achieved by incorporating the proposed term across various batch sizes. The proposed auxiliary loss consistently improves the model performance.

in Def. 5.7. The hyperparameter $\lambda$ for the proposed loss is tuned over $\{0.1, 0.3, 1, 3, 10, 30, 100\}$. Unless otherwise specified, the other settings follow those in Sec. 6.1.

**Performance Gains from Variance Reduction.** The quality of the representations learned through CL is known to be sensitive to the choice of the temperature parameter in contrastive losses, as it influences the distribution of similarities among embeddings (Wang & Liu, 2021). To investigate this sensitivity, we train models using the SimCLR loss with temperature values ranging from 0.07 to 2.00. We compare the standard SimCLR loss against our proposed variant, which incorporates the auxiliary loss introduced in Def. 5.7 to reduce the variance of negative-pair similarities. As shown in Figure 3, incorporating the proposed term leads to a consistently higher and more stable classification accuracy across all temperature settings, alleviating the need for careful temperature tuning.

We further evaluate the auxiliary loss in Def. 5.7 on existing CL methods, including DCL and DHEL. Experiments are conducted on the CIFAR datasets with various batch sizes. As shown in Figure 4, incorporating the proposed term leads to improved classification accuracy, with the effect being more pronounced at smaller batch sizes. Additional results on the ImageNet dataset are presented in Appendix E.

**Caveats of Variance Reduction.** While the auxiliary loss proposed in Def. 5.7 effectively reduces the variance of negative-pair similarities, this reduction can influence both desirable and undesirable sources of variance. On the positive side, it helps reduce variance introduced by mini-batch sampling, which can degrade the representation quality. However, it may also suppress the variance that captures meaningful structure in the data. Further discussion of the limitations of the proposed loss is provided in Appendix F.

## 7. Conclusion

To understand contrastive learning (CL), we mathematically analyze the distributions of similarities of embeddings, measured for positive pairs and negative pairs. Our theoretical results in full-batch settings demonstrate that misalignment of positive pairs becomes inevitable when the average similarity of negative pairs falls below its optimal value, a situation that can arise with existing contrastive losses. In mini-batch settings, we prove that the variance of negative-pair similarities increases as the batch size decreases—a distinctive characteristic absent in full-batch settings and a potential contributor to the performance degradation observed in mini-batch settings. To address this, we propose an auxiliary loss that explicitly reduces the variance of negative-pair similarities. Empirical results show that incorporating the proposed loss improves the performance of CL methods, especially in small-batch settings.

Promising directions for future work include extending this work in two directions. First, disentangling the variance of negative-pair similarities that reflects intrinsic data structure from that caused by mini-batching could enable targeting only the variance that degrades representation quality. Second, analyzing the behavior of embedding similarities not only during pretraining but also during fine-tuning may provide deeper insights into its dynamics throughout different stages of training.

## Acknowledgements

This work was partially supported by the National Research Foundation of Korea (NRF) grant funded by the Ministry of Science and ICT (MSIT) of the Korean government (RS-2024-00341749, RS-2024-00345351, RS-2024-00408003), under the ICT Challenge and Advanced Network of HRD (ICAN) support program (RS-2023-00259934, RS-2025-02283048), supervised by the Institute for Information & Communications Technology Planning & Evaluation (IITP). This research was also supported by the Yonsei University Research Fund (2025-22-0025).

We sincerely thank the anonymous reviewers for their critical reading and constructive feedback enhancing this paper.

## Impact Statement

This paper presents work whose goal is to theoretically understand CL through embedding similarities. There are many potential societal consequences of our work, none which we feel must be specifically highlighted here.

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

# A. Contrastive Losses

We outline how various losses commonly used in CL can be instantiated by the general formulation provided in Def. 3.1 and Def. 3.2. For each case, we specify the corresponding choices of functions and parameters.

## A.1. Contrastive Losses Following Def. 3.1

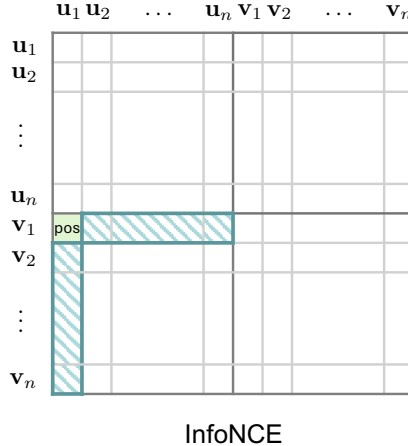

InfoNCE

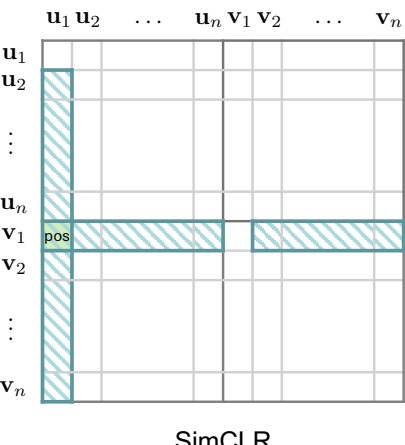

SimCLR

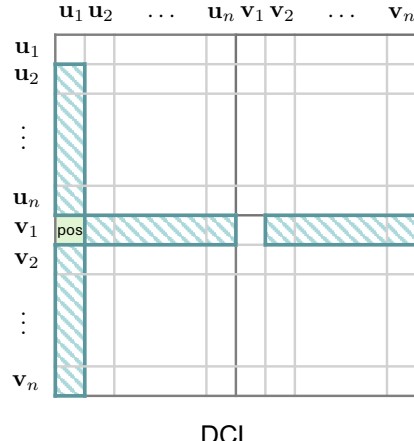

DCL

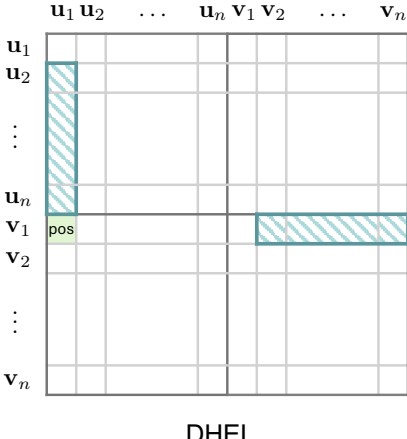

DHEL

*Figure 5.* Illustration comparing four different contrastive losses, all following the form of Def. 3.1. The green area represents the positive pair, while the blue-striped regions indicate the negative pairs that are normalized together with the positive pair for each loss.

*Table 2.* Function and parameter selections in Def. 3.1 that correspond to contrastive losses.

|  | $\phi(x)$ | $\psi(x)$ | $c_1$ | $c_2$ |
|---|---|---|---|---|
| InfoNCE (Oord et al., 2018) | $\exp(x/t)$ | $\log(1+x)$ | 1 | 0 |
| SimCLR (Chen et al., 2020) | $\exp(x/t)$ | $\log(1+x)$ | 1 | 1 |
| DCL (Yeh et al., 2022) | $\exp(x/t)$ | $\log(x)$ | 1 | 1 |
| DHEL (Koromilas et al., 2024) | $\exp(x/t)$ | $\log(x)$ | 0 | 1 |

1. InfoNCE (Oord et al., 2018), CLIP (Radford et al., 2021):

$$
\mathcal{L}_{\text{InfoNCE}}(\boldsymbol{U}, \boldsymbol{V}) = -\frac{1}{2n} \sum_{i \in [n]} \log \left( \frac{\exp\left(\boldsymbol{u}_i^\top \boldsymbol{v}_i / t\right)}{\sum_{j \in [n]} \exp\left(\boldsymbol{u}_i^\top \boldsymbol{v}_j / t\right)} \right) - \frac{1}{2n} \sum_{i \in [n]} \log \left( \frac{\exp\left(\boldsymbol{u}_i^\top \boldsymbol{v}_i / t\right)}{\sum_{j \in [n]} \exp\left(\boldsymbol{u}_j^\top \boldsymbol{v}_i / t\right)} \right) \tag{13}
$$

$$
= \frac{1}{2n} \sum_{i \in [n]} \log \left( 1 + \sum_{j \in [n] \setminus \{i\}} \exp\left((\boldsymbol{v}_j - \boldsymbol{v}_i)^\top \boldsymbol{u}_i / t\right) \right)
$$

$$
+ \frac{1}{2n} \sum_{i \in [n]} \log \left( 1 + \sum_{j \in [n] \setminus \{i\}} \exp\left((\boldsymbol{u}_j - \boldsymbol{u}_i)^\top \boldsymbol{v}_i / t\right) \right),
$$

where $t > 0$ is the temperature parameter.

2. SimCLR (Chen et al., 2020):

$$
\mathcal{L}_{\text{SimCLR}}(\boldsymbol{U}, \boldsymbol{V}) = -\frac{1}{2n} \sum_{i \in [n]} \log \left( \frac{\exp\left(\boldsymbol{u}_i^\top \boldsymbol{v}_i / t\right)}{\sum_{j \in [n]} \exp\left(\boldsymbol{u}_i^\top \boldsymbol{v}_j / t\right) + \sum_{j \in [n] \setminus \{i\}} \exp\left(\boldsymbol{u}_i^\top \boldsymbol{u}_j / t\right)} \right)
$$

$$
- \frac{1}{2n} \sum_{i \in [n]} \log \left( \frac{\exp\left(\boldsymbol{u}_i^\top \boldsymbol{v}_i / t\right)}{\sum_{j \in [n]} \exp\left(\boldsymbol{u}_j^\top \boldsymbol{v}_i / t\right) + \sum_{j \in [n] \setminus \{i\}} \exp\left(\boldsymbol{v}_j^\top \boldsymbol{v}_i / t\right)} \right)
$$

$$
= \frac{1}{2n} \sum_{i \in [n]} \log \left( 1 + \sum_{j \in [n] \setminus \{i\}} \exp\left((\boldsymbol{v}_j - \boldsymbol{v}_i)^\top \boldsymbol{u}_i / t\right) + \sum_{j \in [n] \setminus \{i\}} \exp\left((\boldsymbol{u}_j - \boldsymbol{v}_i)^\top \boldsymbol{u}_i / t\right) \right)
$$

$$
+ \frac{1}{2n} \sum_{i \in [n]} \log \left( 1 + \sum_{j \in [n] \setminus \{i\}} \exp\left((\boldsymbol{u}_j - \boldsymbol{u}_i)^\top \boldsymbol{v}_i / t\right) + \sum_{j \in [n] \setminus \{i\}} \exp\left((\boldsymbol{v}_j - \boldsymbol{u}_i)^\top \boldsymbol{v}_i / t\right) \right),
$$

where $t > 0$ is the temperature parameter.

3. DCL (Yeh et al., 2022):

$$
\mathcal{L}_{\text{DCL}}(\boldsymbol{U}, \boldsymbol{V}) = -\frac{1}{2n} \sum_{i \in [n]} \log \left( \frac{\exp\left(\boldsymbol{u}_i^\top \boldsymbol{v}_i / t\right)}{\sum_{j \in [n] \setminus \{i\}} \exp\left(\boldsymbol{u}_i^\top \boldsymbol{v}_j / t\right) + \sum_{j \in [n] \setminus \{i\}} \exp\left(\boldsymbol{u}_i^\top \boldsymbol{u}_j / t\right)} \right)
$$

$$
- \frac{1}{2n} \sum_{i \in [n]} \log \left( \frac{\exp\left(\boldsymbol{u}_i^\top \boldsymbol{v}_i / t\right)}{\sum_{j \in [n] \setminus \{i\}} \exp\left(\boldsymbol{u}_j^\top \boldsymbol{v}_i / t\right) + \sum_{j \in [n] \setminus \{i\}} \exp\left(\boldsymbol{u}_j^\top \boldsymbol{u}_i / t\right)} \right)
$$

$$
= \frac{1}{2n} \sum_{i \in [n]} \log \left( \sum_{j \in [n] \setminus \{i\}} \left( \exp\left((\boldsymbol{v}_j - \boldsymbol{v}_i)^\top \boldsymbol{u}_i / t\right) + \exp\left((\boldsymbol{u}_j - \boldsymbol{v}_i)^\top \boldsymbol{u}_i / t\right) \right) \right)
$$

$$
+ \frac{1}{2n} \sum_{i \in [n]} \log \left( \sum_{j \in [n] \setminus \{i\}} \left( \exp\left((\boldsymbol{u}_j - \boldsymbol{u}_i)^\top \boldsymbol{v}_i / t\right) + \exp\left((\boldsymbol{v}_j - \boldsymbol{u}_i)^\top \boldsymbol{v}_i / t\right) \right) \right),
$$

where $t > 0$ is the temperature parameter.

4. DHEL (Koromilas et al., 2024):

$$
\mathcal{L}_{\text{DHEL}}(\boldsymbol{U}, \boldsymbol{V}) = -\frac{1}{2n} \sum_{i \in [n]} \log \left( \frac{\exp\left(\boldsymbol{u}_i^\top \boldsymbol{v}_i / t\right)}{\sum_{j \in [n] \setminus \{i\}} \exp\left(\boldsymbol{u}_i^\top \boldsymbol{u}_j / t\right)} \right) - \frac{1}{2n} \sum_{i \in [n]} \log \left( \frac{\exp\left(\boldsymbol{u}_i^\top \boldsymbol{v}_i / t\right)}{\sum_{j \in [n] \setminus \{i\}} \exp\left(\boldsymbol{v}_j^\top \boldsymbol{v}_i / t\right)} \right)
$$

$$
= \frac{1}{2n} \sum_{i \in [n]} \log \left( \sum_{j \in [n] \setminus \{i\}} \exp\left((\boldsymbol{u}_j - \boldsymbol{v}_i)^\top \boldsymbol{u}_i / t\right) \right) + \frac{1}{2n} \sum_{i \in [n]} \log \left( \sum_{j \in [n] \setminus \{i\}} \exp\left((\boldsymbol{v}_j - \boldsymbol{u}_i)^\top \boldsymbol{v}_i / t\right) \right),
$$

where $t > 0$ is the temperature parameter.

## A.2. Contrastive Losses Following Def. 3.2

*Table 3.* Function and parameter selections in Def. 3.2 that correspond to contrastive losses.

| | $\phi(x)$ | $\psi(x)$ | $c_1$ | $c_2$ |
|---|---|---|---|---|
| SigLIP (Zhai et al., 2023) | $-\log(1+\exp(-tx+b))$ | $(n-1)\cdot\log(1+\exp(tx-b))$ | 1 | 0 |
| Spectral Contrastive Loss (HaoChen et al., 2021) | $x$ | $x^2$ | 1 | 0 |

1. SigLIP (Zhai et al., 2023) :

$$\mathcal{L}_{\text{SigLIP}}(\boldsymbol{U}, \boldsymbol{V}) = \frac{1}{n} \sum_{i\in[n]} \log\left(1+\exp\left(-t\boldsymbol{u}_i^\top \boldsymbol{v}_i + b\right)\right) + \frac{1}{n} \sum_{i\in[n]}\sum_{j\in[n]\backslash\{i\}} \log\left(1 + \exp\left(t\boldsymbol{u}_i^\top \boldsymbol{v}_j - b\right)\right)$$

$$= -\frac{1}{n} \sum_{i\in[n]} \left(-\log\left(1+\exp\left(-t\boldsymbol{u}_i^\top \boldsymbol{v}_i + b\right)\right)\right) + \frac{1}{n(n-1)} \sum_{i\in[n]}\sum_{j\in[n]\backslash\{i\}} (n-1)\cdot\log\left(1+\exp\left(t\boldsymbol{u}_i^\top \boldsymbol{v}_j - b\right)\right),$$

where $t > 0$ is the temperature parameter and $b \in \mathbb{R}$ is the bias term.

2. Spectral Contrastive Loss (HaoChen et al., 2021) :

$$\mathcal{L}_{\text{Spectral}}(\boldsymbol{U}, \boldsymbol{V}) = -\frac{1}{n} \sum_{i\in[n]} \boldsymbol{u}_i^\top \boldsymbol{v}_j + \frac{1}{n(n-1)} \sum_{i\in[n]}\sum_{j\in[n]\backslash\{i\}} \left(\boldsymbol{u}_i^\top \boldsymbol{v}_j\right)^2.$$

## B. Distinction Between Cross-View and Within-View Negative Pairs

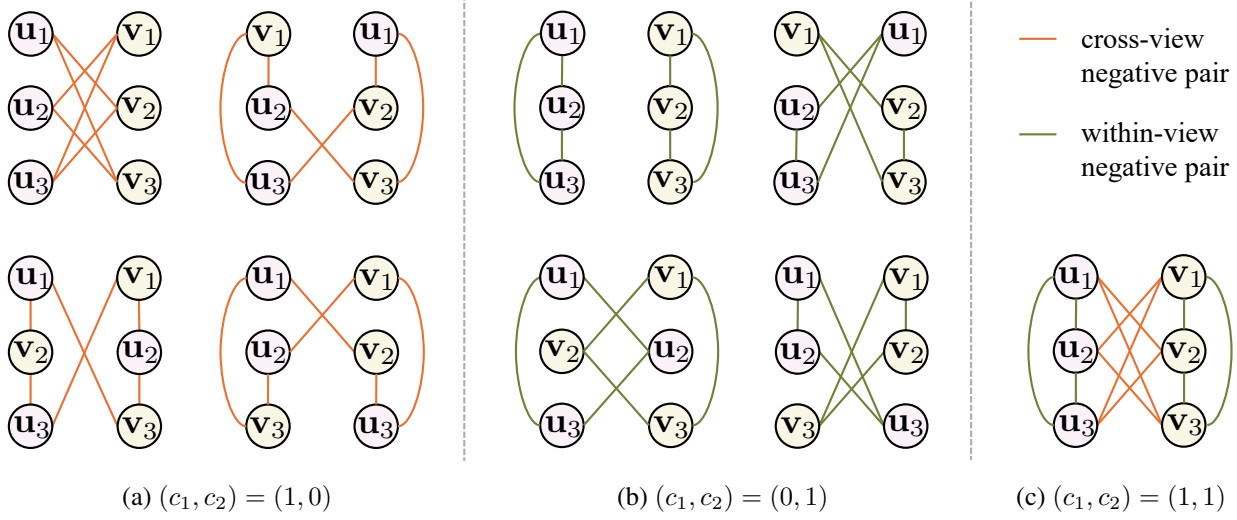

(a) $(c_1, c_2) = (1, 0)$      (b) $(c_1, c_2) = (0, 1)$      (c) $(c_1, c_2) = (1, 1)$

*Figure 6.* Graphs illustrating different loss configurations for $n = 3$.

The key distinction between cross-view and within-view negative pairs in our analysis lies in their structural incorporation within the contrastive loss, rather than in the manner of their generation. To demonstrate that cross-view and within-view negatives are not equivalent, we present a graph-based representation in Figure 6, which reframes Figure 1. In these graphs, each node corresponds to an embedding, and each edge indicates a negative pair considered in the loss.

In the unimodal CL, the distinction between the two views, $u_i$ and $v_i$ for $i \in [3]$, is not semantically meaningful. Thus, $u_i$ and $v_i$ may be interchanged without affecting the results. This implies that the four graphs depicted in Figure 6a are equivalent under permutation of views, and the same reasoning applies to Figure 6b. Nevertheless, the overall graph structures in Figure 6a and Figure 6b remain fundamentally different. One can confirm that cross-view graphs are fully connected bipartite, whereas within-view graphs consist of disconnected subgraphs. This topological difference highlights their non-equivalence.

# C. Proofs

## C.1. Approximation of Uniformity Metric

Under the normality assumption on $\boldsymbol{u}^\top \boldsymbol{v}$, Proposition C.1 gives

$$\log \mathbb{E}\left[\exp\left(2\boldsymbol{u}^\top \boldsymbol{v}\right)\right] = 2\left(\mathbb{E}\left[\boldsymbol{u}^\top \boldsymbol{v}\right] + \mathrm{Var}\left[\boldsymbol{u}^\top \boldsymbol{v}\right]\right).$$

Accordingly, the uniformity metric can be approximated as

$$\log \mathbb{E}_{\boldsymbol{u}\sim f_\sharp\hat{p}_x\ \boldsymbol{v}\sim f\sharp\hat{p}_y}\left[\exp\left(-\|\boldsymbol{u}-\boldsymbol{v}\|_2^2\right)\right] \approx \log \mathbb{E}_{(\boldsymbol{u},\boldsymbol{v})\sim f\sharp\hat{p}_{\mathrm{neg}}}\left[\exp\left(-\|\boldsymbol{u}-\boldsymbol{v}\|_2^2\right)\right]$$
$$\approx 2\left(\mathbb{E}\left[s(f;\hat{p}_{\mathrm{neg}})\right] + \mathrm{Var}\left[s(f;\hat{p}_{\mathrm{neg}})\right] - 1\right),$$

by Proposition C.2, as $n$ goes to infinity.

**Proposition C.1.** *Assume that the random variable $\boldsymbol{u}^\top \boldsymbol{v}$ follows the normal distribution. Then,*

$$\log \mathbb{E}\left[\exp\left(2\boldsymbol{u}^\top \boldsymbol{v}\right)\right] = 2\left(\mathbb{E}\left[\boldsymbol{u}^\top \boldsymbol{v}\right] + \mathrm{Var}\left[\boldsymbol{u}^\top \boldsymbol{v}\right]\right)$$

*Proof.* Let $X = \boldsymbol{u}^\top \boldsymbol{v}$. Since $X$ follows the normal distribution, we define $\mu := \mathbb{E}[X]$ and $\sigma^2 := \mathrm{Var}[X]$.

Note that the moment generating function of normal distribution is given by

$$\mathbb{E}\left[\exp(tX)\right] = \exp\left(\mu t + \frac{\sigma^2 t^2}{2}\right).$$

Substituting $t = 2$, we have

$$\mathbb{E}\left[\exp(2X)\right] = \exp\left(2\mu + 2\sigma^2\right) = \exp\left(2\mathbb{E}[X] + 2\mathrm{Var}[X]\right),$$

which is equal to

$$\log \mathbb{E}\left[\exp(2X)\right] = 2\left(\mathbb{E}[X] + \mathrm{Var}[X]\right).$$

Since $X = \boldsymbol{u}^\top \boldsymbol{v}$, we conclude

$$\log \mathbb{E}\left[\exp\left(2\boldsymbol{u}^\top \boldsymbol{v}\right)\right] = 2\left(\mathbb{E}\left[\boldsymbol{u}^\top \boldsymbol{v}\right] + \mathrm{Var}\left[\boldsymbol{u}^\top \boldsymbol{v}\right]\right).$$

$\square$

## C.2. Proofs for Relation Between Positive and Negative Pairs

**Proposition C.2.** *The distribution of negative pairs satisfies $p_{\mathrm{neg}}(\boldsymbol{x},\boldsymbol{y}) = p_x(\boldsymbol{x})p_y(\boldsymbol{y})$ for all $\boldsymbol{x}$ and $\boldsymbol{y}$. However, for a training dataset of size $n$, the empirical distribution of negative pairs is given by*

$$\hat{p}_{\mathrm{neg}}(\boldsymbol{x},\boldsymbol{y}) = \frac{n}{n-1}\cdot \hat{p}_x(\boldsymbol{x})\hat{p}_y(\boldsymbol{y}) - \frac{1}{n-1}\cdot \hat{p}_{\mathrm{pos}}(\boldsymbol{x},\boldsymbol{y}),$$

*for all $\boldsymbol{x}$ and $\boldsymbol{y}$.*

*Proof.* The empirical distribution of positive pairs, $\hat{p}_{\mathrm{pos}}$, is defined under the assumption that all instances are equally weighted with the probability $\Pr\{\boldsymbol{I} = i\} = \frac{1}{n}$ for all $i \in [n]$, where $\boldsymbol{I}$ is a random variable representing the index. Under this assumption, the probability that a randomly selected pair is positive is

$$\Pr\{\mathrm{pos}\} = \Pr\{\boldsymbol{i} = \boldsymbol{i}'\} = \sum_{i\in[n]}\Pr\{\boldsymbol{i}=i\}\Pr\{\boldsymbol{i}=i\} = n\cdot\frac{1}{n^2} = \frac{1}{n}.$$

Then, the empirical distribution of negative pairs, $\hat{p}_{\mathrm{neg}}$, is subsequently derived as

$$\hat{p}_x(\boldsymbol{x})\hat{p}_y(\boldsymbol{y}) = \hat{p}_{\mathrm{pos}}(\boldsymbol{x},\boldsymbol{y})\Pr\{\mathrm{pos}\} + \hat{p}_{\mathrm{neg}}(\boldsymbol{x},\boldsymbol{y})(1-\Pr\{\mathrm{pos}\})$$
$$= \hat{p}_{\mathrm{pos}}(\boldsymbol{x},\boldsymbol{y})\cdot\frac{1}{n} + \hat{p}_{\mathrm{neg}}(\boldsymbol{x},\boldsymbol{y})\cdot\frac{n-1}{n}, \tag{14}$$

which leads to

$$\hat{p}_{\mathrm{neg}}(\boldsymbol{x},\boldsymbol{y}) = \frac{n}{n-1}\cdot\hat{p}_x(\boldsymbol{x})\hat{p}_y(\boldsymbol{y}) - \frac{1}{n-1}\cdot\hat{p}_{\mathrm{pos}}(\boldsymbol{x},\boldsymbol{y}).$$

Moreover, as $n \to \infty$, the above result implies that $p_{\mathrm{neg}}(\boldsymbol{x},\boldsymbol{y}) = p_x(\boldsymbol{x})p_y(\boldsymbol{y})$. □

**Lemma C.3.** *Assume that the encoder $f(\cdot)$ satisfies $\|f(\boldsymbol{x})\|_2^2 = 1$ for all $\boldsymbol{x}$. For any distribution $p$, the following holds.*

$$1 - \mathbb{E}_{(\boldsymbol{u},\boldsymbol{v})\sim f_\sharp p_{\mathrm{pos}}}\left[\boldsymbol{u}^\top\boldsymbol{v}\right] + \mathbb{E}_{\substack{\boldsymbol{u}\sim f_\sharp p_x\\\boldsymbol{v}\sim f_\sharp p_y}}\left[\boldsymbol{u}^\top\boldsymbol{v}\right] = \frac{1}{2}\operatorname{tr}\left(\operatorname{Var}_{(\boldsymbol{u},\boldsymbol{v})\sim f_\sharp p_{\mathrm{pos}}}[\boldsymbol{u}-\boldsymbol{v}]\right) + \frac{1}{2}\left\|\mathbb{E}_{(\boldsymbol{u},\boldsymbol{v})\sim f_\sharp p_{\mathrm{pos}}}\left[\boldsymbol{u}+\boldsymbol{v}\right]\right\|_2^2.$$

*Proof.* Note that

$$\left\|\mathbb{E}_{(\boldsymbol{u},\boldsymbol{v})\sim f_\sharp p_{\mathrm{pos}}}\left[\boldsymbol{u}-\boldsymbol{v}\right]\right\|_2^2 - \left\|\mathbb{E}_{(\boldsymbol{u},\boldsymbol{v})\sim f_\sharp p_{\mathrm{pos}}}\left[\boldsymbol{u}+\boldsymbol{v}\right]\right\|_2^2 = -4\mathbb{E}_{(\boldsymbol{u},\boldsymbol{v})\sim f_\sharp p_{\mathrm{pos}}}\left[\boldsymbol{u}\right]^\top\mathbb{E}_{(\boldsymbol{u},\boldsymbol{v})\sim f_\sharp p_{\mathrm{pos}}}\left[\boldsymbol{v}\right]$$

$$= -4\mathbb{E}_{\boldsymbol{u}\sim f_\sharp p_x}\left[\boldsymbol{u}\right]^\top\mathbb{E}_{\boldsymbol{v}\sim f_\sharp p_y}\left[\boldsymbol{v}\right], \tag{15}$$

where the equality in (15) follows from the assumption of matching marginals.

From the definition of variance, we have

$$\operatorname{tr}\left(\operatorname{Var}_{(\boldsymbol{u},\boldsymbol{v})\sim f_\sharp p_{\mathrm{pos}}}[\boldsymbol{u}-\boldsymbol{v}]\right) = \mathbb{E}_{(\boldsymbol{u},\boldsymbol{v})\sim f_\sharp p_{\mathrm{pos}}}\left[\operatorname{tr}\left(\left((\boldsymbol{u}-\boldsymbol{v}) - \mathbb{E}_{(\boldsymbol{u},\boldsymbol{v})\sim f_\sharp p_{\mathrm{pos}}}\left[\boldsymbol{u}-\boldsymbol{v}\right]\right)\left((\boldsymbol{u}-\boldsymbol{v}) - \mathbb{E}_{(\boldsymbol{u},\boldsymbol{v})\sim f_\sharp p_{\mathrm{pos}}}\left[\boldsymbol{u}-\boldsymbol{v}\right]\right)^\top\right)\right]$$

$$= \mathbb{E}_{(\boldsymbol{u},\boldsymbol{v})\sim f_\sharp p_{\mathrm{pos}}}\left[\left\|(\boldsymbol{u}-\boldsymbol{v}) - \mathbb{E}_{(\boldsymbol{u},\boldsymbol{v})\sim f_\sharp p_{\mathrm{pos}}}\left[\boldsymbol{u}-\boldsymbol{v}\right]\right\|_2^2\right]$$

$$= \mathbb{E}_{(\boldsymbol{u},\boldsymbol{v})\sim f_\sharp p_{\mathrm{pos}}}\left[\left\|\boldsymbol{u}-\boldsymbol{v}\right\|_2^2\right] - \left\|\mathbb{E}_{(\boldsymbol{u},\boldsymbol{v})\sim f_\sharp p_{\mathrm{pos}}}\left[\boldsymbol{u}-\boldsymbol{v}\right]\right\|_2^2$$

$$= \mathbb{E}_{(\boldsymbol{u},\boldsymbol{v})\sim f_\sharp p_{\mathrm{pos}}}\left[2 - 2\cdot\boldsymbol{u}^\top\boldsymbol{v}\right] - \left\|\mathbb{E}_{(\boldsymbol{u},\boldsymbol{v})\sim f_\sharp p_{\mathrm{pos}}}\left[\boldsymbol{u}+\boldsymbol{v}\right]\right\|_2^2 + 4\mathbb{E}_{\boldsymbol{u}\sim f_\sharp p_x}\left[\boldsymbol{u}\right]^\top\mathbb{E}_{\boldsymbol{v}\sim f_\sharp p_y}\left[\boldsymbol{v}\right]$$

$$= 2 - 2\mathbb{E}_{(\boldsymbol{u},\boldsymbol{v})\sim f_\sharp p_{\mathrm{pos}}}\left[\boldsymbol{u}^\top\boldsymbol{v}\right] - \left\|\mathbb{E}_{(\boldsymbol{u},\boldsymbol{v})\sim f_\sharp p_{\mathrm{pos}}}\left[\boldsymbol{u}+\boldsymbol{v}\right]\right\|_2^2 + 4\mathbb{E}_{\boldsymbol{u}\sim f_\sharp p_x}\left[\boldsymbol{u}\right]^\top\mathbb{E}_{\boldsymbol{v}\sim f_\sharp p_y}\left[\boldsymbol{v}\right]. \tag{16}$$

By rearranging (16) and dividing by 2, we have

$$\frac{1}{2}\operatorname{tr}\left(\operatorname{Var}_{(\boldsymbol{u},\boldsymbol{v})\sim f_\sharp p_{\mathrm{pos}}}[\boldsymbol{u}-\boldsymbol{v}]\right) + \left\|\mathbb{E}_{(\boldsymbol{u},\boldsymbol{v})\sim f_\sharp p_{\mathrm{pos}}}\left[\boldsymbol{u}+\boldsymbol{v}\right]\right\|_2^2 = 1 - \mathbb{E}_{(\boldsymbol{u},\boldsymbol{v})\sim f_\sharp p_{\mathrm{pos}}}\left[\boldsymbol{u}^\top\boldsymbol{v}\right] + \mathbb{E}_{\boldsymbol{u}\sim f_\sharp p_x}\left[\boldsymbol{u}\right]^\top\mathbb{E}_{\boldsymbol{v}\sim f_\sharp p_y}\left[\boldsymbol{v}\right]$$

$$= 1 - \mathbb{E}_{(\boldsymbol{u},\boldsymbol{v})\sim f_\sharp p_{\mathrm{pos}}}\left[\boldsymbol{u}^\top\boldsymbol{v}\right] + \mathbb{E}_{\substack{\boldsymbol{u}\sim f_\sharp p_x\\\boldsymbol{v}\sim f_\sharp p_y}}\left[\boldsymbol{u}^\top\boldsymbol{v}\right]$$

□

**Lemma C.4.** *Assume that the encoder $f(\cdot)$ satisfies $\|f(\boldsymbol{x})\|_2^2 = 1$ for all $\boldsymbol{x}$. For any empirical distribution $\hat{p}$ with a sample size of $n$, the following holds.*

$$1 - \frac{n-1}{n}\mathbb{E}_{(\boldsymbol{u},\boldsymbol{v})\sim f_\sharp\hat{p}_{\mathrm{pos}}}\left[\boldsymbol{u}^\top\boldsymbol{v}\right] + \frac{n-1}{n}\mathbb{E}_{(\boldsymbol{u},\boldsymbol{v})\sim f_\sharp\hat{p}_{\mathrm{neg}}}\left[\boldsymbol{u}^\top\boldsymbol{v}\right] = \frac{1}{2}\operatorname{tr}\left(\operatorname{Var}_{(\boldsymbol{u},\boldsymbol{v})\sim f_\sharp\hat{p}_{\mathrm{pos}}}[\boldsymbol{u}-\boldsymbol{v}]\right) + \frac{1}{2}\left\|\mathbb{E}_{(\boldsymbol{u},\boldsymbol{v})\sim f_\sharp\hat{p}_{\mathrm{pos}}}\left[\boldsymbol{u}+\boldsymbol{v}\right]\right\|_2^2.$$

*Proof.* Using Proposition C.2, the expectation over the empirical distribution can be decomposed into the expectations of positive and negative pairs as follows.

$$\mathbb{E}_{\substack{\boldsymbol{u}\sim f_\sharp\hat{p}_x\\\boldsymbol{v}\sim f_\sharp\hat{p}_y}}\left[\boldsymbol{u}^\top\boldsymbol{v}\right] = \frac{1}{n}\mathbb{E}_{(\boldsymbol{u},\boldsymbol{v})\sim f_\sharp\hat{p}_{\mathrm{pos}}}\left[\boldsymbol{u}^\top\boldsymbol{v}\right] + \frac{n-1}{n}\mathbb{E}_{(\boldsymbol{u},\boldsymbol{v})\sim f_\sharp\hat{p}_{\mathrm{neg}}}\left[\boldsymbol{u}^\top\boldsymbol{v}\right].$$

Applying Lemma C.3 to the empirical distribution $\hat{p}$, we have

$$\frac{1}{2}\operatorname{tr}\left(\operatorname{Var}_{(\boldsymbol{u},\boldsymbol{v})\sim f_\sharp\hat{p}_{\mathrm{pos}}}[\boldsymbol{u}-\boldsymbol{v}]\right) + \left\|\mathbb{E}_{(\boldsymbol{u},\boldsymbol{v})\sim f_\sharp\hat{p}_{\mathrm{pos}}}\left[\boldsymbol{u}+\boldsymbol{v}\right]\right\|_2^2 = 1 - \mathbb{E}_{(\boldsymbol{u},\boldsymbol{v})\sim f_\sharp\hat{p}_{\mathrm{pos}}}\left[\boldsymbol{u}^\top\boldsymbol{v}\right] + \mathbb{E}_{\substack{\boldsymbol{u}\sim f_\sharp\hat{p}_x\\\boldsymbol{v}\sim f_\sharp p_y}}\left[\boldsymbol{u}^\top\boldsymbol{v}\right]$$

$$= 1 - \frac{n-1}{n}\mathbb{E}_{(\boldsymbol{u},\boldsymbol{v})\sim f_\sharp\hat{p}_{\mathrm{pos}}}\left[\boldsymbol{u}^\top\boldsymbol{v}\right] + \frac{n-1}{n}\mathbb{E}_{(\boldsymbol{u},\boldsymbol{v})\sim f_\sharp\hat{p}_{\mathrm{neg}}}\left[\boldsymbol{u}^\top\boldsymbol{v}\right].$$

□

**Theorem C.5.** *Assume that the encoder $f(\cdot)$ satisfies $\|f(\boldsymbol{x})\|_2^2 = 1$ for all $\boldsymbol{x}$. For any empirical distribution $\hat{p}$ with a sample size of $n$, the following inequality holds.*

$$\mathbb{E}_{(\boldsymbol{u},\boldsymbol{v})\sim f_\sharp \hat{p}_{\mathrm{pos}}} \left[\boldsymbol{u}^\top \boldsymbol{v}\right] \leq 1 + \left(\mathbb{E}_{(\boldsymbol{u},\boldsymbol{v})\sim f_\sharp \hat{p}_{\mathrm{neg}}} \left[\boldsymbol{u}^\top \boldsymbol{v}\right] + \frac{1}{n-1}\right),$$

*where equality holds if and only if $\mathrm{tr}\left(\mathrm{Var}_{(\boldsymbol{u},\boldsymbol{v})\sim f_\sharp \hat{p}_{\mathrm{pos}}}[\boldsymbol{u}-\boldsymbol{v}]\right) = 0$ and $\mathbb{E}_{\boldsymbol{u}\sim f_\sharp \hat{p}_x}[\boldsymbol{u}] + \mathbb{E}_{\boldsymbol{v}\sim f_\sharp \hat{p}_y}[\boldsymbol{v}] = \boldsymbol{0}$.*

*Proof.* From Lemma C.4, and variance and norm are non-negative, we have

$$\begin{aligned}
\mathbb{E}_{(\boldsymbol{u},\boldsymbol{v})\sim f_\sharp \hat{p}_{\mathrm{pos}}} \left[\boldsymbol{u}^\top \boldsymbol{v}\right] &= \frac{n}{n-1} + \mathbb{E}_{(\boldsymbol{u},\boldsymbol{v})\sim f_\sharp \hat{p}_{\mathrm{neg}}} \left[\boldsymbol{u}^\top \boldsymbol{v}\right] \\
&\quad - \frac{n}{2(n-1)} \mathrm{tr}\left(\mathrm{Var}_{(\boldsymbol{u},\boldsymbol{v})\sim f_\sharp \hat{p}_{\mathrm{pos}}}[\boldsymbol{u}-\boldsymbol{v}]\right) - \frac{n}{2(n-1)} \left\|\mathbb{E}_{(\boldsymbol{u},\boldsymbol{v})\sim f_\sharp \hat{p}_{\mathrm{pos}}}[\boldsymbol{u}+\boldsymbol{v}]\right\|_2^2 \\
&\leq \frac{n}{n-1} + \mathbb{E}_{(\boldsymbol{u},\boldsymbol{v})\sim f_\sharp \hat{p}_{\mathrm{neg}}} \left[\boldsymbol{u}^\top \boldsymbol{v}\right],
\end{aligned} \tag{17}$$

where equality in (17) holds if and only if $\mathrm{tr}\left(\mathrm{Var}_{(\boldsymbol{u},\boldsymbol{v})\sim f_\sharp \hat{p}_{\mathrm{pos}}}[\boldsymbol{u}-\boldsymbol{v}]\right) = 0$ and $\left\|\mathbb{E}_{(\boldsymbol{u},\boldsymbol{v})\sim f_\sharp \hat{p}_{\mathrm{pos}}}[\boldsymbol{u}+\boldsymbol{v}]\right\|_2^2 = 0$. Moreover, the condition of $\left\|\mathbb{E}_{(\boldsymbol{u},\boldsymbol{v})\sim f_\sharp \hat{p}_{\mathrm{pos}}}[\boldsymbol{u}+\boldsymbol{v}]\right\|_2^2 = 0$ is equal to $\mathbb{E}_{\boldsymbol{u}\sim f_\sharp \hat{p}_x}[\boldsymbol{u}] + \mathbb{E}_{\boldsymbol{v}\sim f_\sharp \hat{p}_y}[\boldsymbol{v}] = \boldsymbol{0}$. As a result, the following holds:

$$\mathbb{E}_{(\boldsymbol{u},\boldsymbol{v})\sim f_\sharp \hat{p}_{\mathrm{pos}}} \left[\boldsymbol{u}^\top \boldsymbol{v}\right] \leq 1 + \left(\mathbb{E}_{(\boldsymbol{u},\boldsymbol{v})\sim f_\sharp \hat{p}_{\mathrm{neg}}} \left[\boldsymbol{u}^\top \boldsymbol{v}\right] + \frac{1}{n-1}\right).$$

$\square$

## C.3. Proofs for Full-Batch CL

**Lemma C.6** (Restatement of Lemma 1 in Lee et al. (2024)). *Let $\boldsymbol{u}_1, \boldsymbol{v}_1, \boldsymbol{u}_2, \boldsymbol{v}_2, \cdots \boldsymbol{u}_n, \boldsymbol{v}_n$ be $2n$ vectors, satisfying $\boldsymbol{u}_i^\top \boldsymbol{u}_i = \boldsymbol{v}_i^\top \boldsymbol{v}_i = 1$ for all $i \in [n]$. Then, the following inequality holds.*

$$\frac{1}{n(n-1)} \sum_{i\in[n]} \sum_{j\in[n]\setminus\{i\}} \boldsymbol{u}_i^\top \boldsymbol{v}_j \geq \frac{n-2}{2n(n-1)} \sum_{i\in[n]} \boldsymbol{u}_i^\top \boldsymbol{v}_i - \frac{n}{2(n-1)}, \tag{18}$$

*where the equality conditions are*

$$\begin{cases} \boldsymbol{u}_i - \boldsymbol{v}_i = \boldsymbol{c} \text{ for all } i \in [n], \text{ for some constant vector } \boldsymbol{c}, \\ \sum_{i\in[n]} \boldsymbol{u}_i + \sum_{i\in[n]} \boldsymbol{v}_i = \boldsymbol{0}. \end{cases}$$

*Proof.* By using Jensen's inequality, we have

$$\begin{aligned}
\frac{1}{n} \sum_{i\in[n]} \|\boldsymbol{u}_i - \boldsymbol{v}_i\|_2^2 &\geq \left\|\frac{1}{n} \sum_{i\in[n]} (\boldsymbol{u}_i - \boldsymbol{v}_i)\right\|_2^2 \\
&= \left\|\frac{1}{n} \sum_{i\in[n]} (\boldsymbol{u}_i + \boldsymbol{v}_i)\right\|_2^2 - 4 \left(\frac{1}{n} \sum_{i\in[n]} \boldsymbol{u}_i\right)^\top \left(\frac{1}{n} \sum_{i\in[n]} \boldsymbol{v}_i\right) \\
&\geq -4 \left(\frac{1}{n} \sum_{i\in[n]} \boldsymbol{u}_i\right)^\top \left(\frac{1}{n} \sum_{i\in[n]} \boldsymbol{v}_i\right),
\end{aligned}$$

where the equality conditions are

$$\begin{cases} \boldsymbol{u}_i - \boldsymbol{v}_i = \boldsymbol{c} \text{ for all } i \in [n], \text{ for some constant vector } \boldsymbol{c}, \\ \sum_{i\in[n]} \boldsymbol{u}_i + \sum_{i\in[n]} \boldsymbol{v}_i = \boldsymbol{0}. \end{cases} \tag{19}$$

From the above, it follows that

$$\left(\frac{1}{n}\sum_{i\in[n]}\boldsymbol{u}_i\right)^{\top}\left(\frac{1}{n}\sum_{i\in[n]}\boldsymbol{v}_i\right) \geq -\frac{1}{4n}\sum_{i\in[n]}\|\boldsymbol{u}_i-\boldsymbol{v}_i\|_2^2 \tag{20}$$

$$= -\frac{1}{2}+\frac{1}{2n}\sum_{i\in[n]}\boldsymbol{u}_i^{\top}\boldsymbol{v}_i, \tag{21}$$

where the last equality uses $\|\boldsymbol{u}_i-\boldsymbol{v}_i\|_2^2 = 2-2\boldsymbol{u}_i^{\top}\boldsymbol{v}_i$ for all $i\in[n]$.

Note that the inner product of centroids is

$$\left(\frac{1}{n}\sum_{i\in[n]}\boldsymbol{u}_i\right)^{\top}\left(\frac{1}{n}\sum_{i\in[n]}\boldsymbol{v}_i\right) = \frac{1}{n^2}\sum_{i\in[n]}\boldsymbol{u}_i^{\top}\boldsymbol{v}_i + \frac{1}{n^2}\sum_{i\in[n]}\sum_{j\in[n]\setminus\{i\}}\boldsymbol{u}_i^{\top}\boldsymbol{v}_j.$$

Combining this with (21), we have

$$\frac{1}{n^2}\sum_{i\in[n]}\sum_{j\in[n]\setminus\{i\}}\boldsymbol{u}_i^{\top}\boldsymbol{v}_j = \left(\frac{1}{n}\sum_{i\in[n]}\boldsymbol{u}_i\right)^{\top}\left(\frac{1}{n}\sum_{i\in[n]}\boldsymbol{v}_i\right) - \frac{1}{n^2}\sum_{i\in[n]}\boldsymbol{u}_i^{\top}\boldsymbol{v}_i$$

$$\geq -\frac{1}{2}+\frac{1}{2n}\sum_{i\in[n]}\boldsymbol{u}_i^{\top}\boldsymbol{v}_i - \frac{1}{n^2}\sum_{i\in[n]}\boldsymbol{u}_i^{\top}\boldsymbol{v}_i$$

$$= \frac{n-2}{2n^2}\sum_{i\in[n]}\boldsymbol{u}_i^{\top}\boldsymbol{v}_i - \frac{1}{2}.$$

The inequality follows from (20), with equality achieved under the conditions specified in (19). □

**Lemma C.7.** *Let $\boldsymbol{u}_1,\boldsymbol{v}_1,\boldsymbol{u}_2,\boldsymbol{v}_2,\cdots\boldsymbol{u}_n,\boldsymbol{v}_n$ be $2n$ vectors, satisfying $\boldsymbol{u}_i^{\top}\boldsymbol{u}_i = \boldsymbol{v}_i^{\top}\boldsymbol{v}_i = 1$ for all $i\in[n]$. Then, the following inequality holds.*

$$\frac{1}{n(n-1)}\sum_{i\in[n]}\sum_{j\in[n]\setminus\{i\}}(\boldsymbol{u}_i^{\top}\boldsymbol{u}_j+\boldsymbol{v}_i^{\top}\boldsymbol{v}_j+2\boldsymbol{u}_i^{\top}\boldsymbol{v}_i) \geq -\frac{2}{n(n-1)}\sum_{i\in[n]}\boldsymbol{u}_i^{\top}\boldsymbol{v}_i - \frac{2}{n-1}, \tag{22}$$

*where equality holds if and only if $\sum_{i\in[n]}(\boldsymbol{u}_i+\boldsymbol{v}_i) = \boldsymbol{0}$.*

*Proof.* Note that

$$\left\|\sum_{i\in[n]}(\boldsymbol{u}_i+\boldsymbol{v}_i)\right\|_2^2 = \left(\sum_{i\in[n]}\boldsymbol{u}_i\right)^{\top}\left(\sum_{i\in[n]}\boldsymbol{u}_i\right) + \left(\sum_{i\in[n]}\boldsymbol{v}_i\right)^{\top}\left(\sum_{i\in[n]}\boldsymbol{v}_i\right) + 2\left(\sum_{i\in[n]}\boldsymbol{u}_i\right)^{\top}\left(\sum_{i\in[n]}\boldsymbol{v}_i\right)$$

$$= n+\sum_{i\in[n]}\sum_{j\in[n]\setminus\{i\}}\boldsymbol{u}_i^{\top}\boldsymbol{u}_j + n+\sum_{i\in[n]}\sum_{j\in[n]\setminus\{i\}}\boldsymbol{v}_i^{\top}\boldsymbol{v}_j + 2\sum_{i\in[n]}\boldsymbol{u}_i^{\top}\boldsymbol{v}_i + 2\sum_{i\in[n]}\sum_{j\in[n]\setminus\{i\}}\boldsymbol{u}_i^{\top}\boldsymbol{v}_j$$

$$= 2n+\sum_{i\in[n]}\sum_{j\in[n]\setminus\{i\}}(\boldsymbol{u}_i^{\top}\boldsymbol{u}_j+\boldsymbol{v}_i^{\top}\boldsymbol{v}_j+2\boldsymbol{u}_i^{\top}\boldsymbol{v}_j) + 2\sum_{i\in[n]}\boldsymbol{u}_i^{\top}\boldsymbol{v}_i.$$

Rearranging terms, we have

$$\frac{1}{n(n-1)}\sum_{i\in[n]}\sum_{j\in[n]\setminus\{i\}}(\boldsymbol{u}_i^{\top}\boldsymbol{u}_j+\boldsymbol{v}_i^{\top}\boldsymbol{v}_j+2\boldsymbol{u}_i^{\top}\boldsymbol{v}_i) = \frac{1}{n(n-1)}\left\|\sum_{i\in[n]}(\boldsymbol{u}_i+\boldsymbol{v}_i)\right\|_2^2 - \frac{2}{n(n-1)}\sum_{i\in[n]}\boldsymbol{u}_i^{\top}\boldsymbol{v}_i - \frac{2}{n-1}$$

$$\geq -\frac{2}{n(n-1)}\sum_{i\in[n]}\boldsymbol{u}_i^{\top}\boldsymbol{v}_i - \frac{2}{n-1},$$

where the equality condition is $\sum_{i\in[n]}(\boldsymbol{u}_i+\boldsymbol{v}_i) = \boldsymbol{0}$. □

**Lemma C.8.** *Let $u_1, v_1, u_2, v_2, \cdots u_n, v_n$ be $2n$ vectors, satisfying $u_i^\top u_i = v_i^\top v_i = 1$ for all $i \in [n]$. Then, the following inequality holds.*

$$\frac{1}{n(n-1)} \sum_{i \in [n]} \sum_{j \in [n] \setminus \{i\}} (u_i^\top u_j + v_i^\top v_j) \geq -\frac{2}{n-1}, \tag{23}$$

*where equality holds if and only if $\sum_{i \in [n]} u_i = \sum_{i \in [n]} v_i = \mathbf{0}$.*

*Proof.* Since $u_i^\top u_i = v_i^\top v_i = 1$ for all $i \in [n]$, we have

$$\frac{1}{n(n-1)} \sum_{i \in [n]} \sum_{j \in [n] \setminus \{i\}} (u_i^\top u_j + v_i^\top v_j) = \frac{1}{n(n-1)} \left( \sum_{i \in [n]} \sum_{j \in [n]} u_i^\top u_j - \sum_{i \in [n]} u_i^\top u_i + \sum_{i \in [n]} \sum_{j \in [n]} v_i^\top v_j - \sum_{i \in [n]} v_i^\top v_i \right)$$

$$= \frac{1}{n(n-1)} \left( \left\| \sum_{i \in [n]} u_i \right\|_2^2 + \left\| \sum_{j \in [n]} v_j \right\|_2^2 \right) - \frac{2}{n-1}$$

$$\geq -\frac{2}{n-1},$$

where the equality condition is $\sum_{i \in [n]} u_i = \sum_{i \in [n]} v_i = \mathbf{0}$. $\qquad \square$

**Theorem C.9.** *Suppose that $d \geq n-1$. Let the contrastive loss $\mathcal{L}(U, V)$ be one of the following forms.*

i. *$\mathcal{L}_{\text{info-sym}}(U, V)$ in Def. 3.1.*

ii. *$\mathcal{L}_{\text{ind-add}}(U, V)$ in Def. 3.2, where $(c_1, c_2) \in \{(0, 1), (1, 1)\}$.*

iii. *$\mathcal{L}_{\text{ind-add}}(U, V)$ in Def. 3.2, where $(c_1, c_2) = (1, 0)$ and $\phi'(1) > \frac{n-2}{2(n-1)} \cdot \psi'\left(-\frac{1}{n-1}\right)$.*

*Then, the embedding similarities for the full-batch optimal encoder $f^\star$ in (1) satisfy*

$$s(f^\star; \hat{p}_{\text{pos}}) = 1, \qquad s(f^\star; \hat{p}_{\text{neg}}) = -\frac{1}{n-1}.$$

**Theorem C.10.** *Suppose that $d \geq n$. Let the contrastive loss $\mathcal{L}(U, V)$ be the form of $\mathcal{L}_{\text{ind-add}}(U, V)$ in Def. 3.2, where $(c_1, c_2) = (1, 0)$ and $\phi'(1) < \frac{n-2}{2(n-1)} \cdot \psi'\left(-\frac{1}{n-1}\right)$. Then, embedding similarities for the full-batch optimal encoder $f^\star$ in (1) satisfy*

$$s(f^\star; \hat{p}_{\text{pos}}) < 1, \qquad s(f^\star; \hat{p}_{\text{neg}}) < -\frac{1}{n-1}.$$

*Proof of Theorem. C.9 and Theorem. C.10.* We prove for each category of loss, $\mathcal{L}_{\text{info-sym}}(U, V)$ in Def. 3.1 and $\mathcal{L}_{\text{ind-add}}(U, V)$ in Def. 3.2, separately.

First, consider the case of $\mathcal{L}_{\text{info-sym}}(U, V)$ in Def. 3.1. By using Jensen's inequality, we have

$$\mathcal{L}_{\text{info}}(U, V) = \frac{1}{n} \sum_{i \in [n]} \psi \left( c_1 \sum_{j \in [n] \setminus \{i\}} \phi\left((v_j - v_i)^\top u_i\right) + c_2 \sum_{j \in [n] \setminus \{i\}} \phi\left((u_j - v_i)^\top u_i\right) \right)$$

$$\geq \frac{1}{n} \sum_{i \in [n]} \psi \left( (c_1 + c_2)(n-1) \cdot \phi\left( \frac{1}{(c_1 + c_2)(n-1)} \sum_{j \in [n] \setminus \{i\}} \left(c_1 (v_j - v_i)^\top u_i + c_2 (u_j - v_i)^\top u_i\right) \right) \right)$$

$$= \frac{1}{n} \sum_{i \in [n]} \psi \left( (c_1 + c_2)(n-1) \cdot \phi\left( \frac{1}{(c_1 + c_2)(n-1)} \sum_{j \in [n] \setminus \{i\}} \left(c_1 v_j^\top u_i + c_2 u_j^\top u_i - (c_1 + c_2) v_i^\top u_i\right) \right) \right), \tag{24}$$

where equality in (24) holds if the argument of $\phi$ is constant for all $j \in [n] \setminus \{i\}$.

Let define $h_{(c_1,c_2)}(x) := \psi((c_1 + c_2)(n-1)\phi(x))$, which is a convex and increasing function. Then, we have

$$
\mathcal{L}_{\text{info-sym}}(\boldsymbol{U}, \boldsymbol{V}) = \frac{1}{2}\mathcal{L}_{\text{info}}(\boldsymbol{U}, \boldsymbol{V}) + \frac{1}{2}\mathcal{L}_{\text{info}}(\boldsymbol{V}, \boldsymbol{U})
$$

$$
\geq \frac{1}{2n} \sum_{i \in [n]} \psi\left((c_1 + c_2)(n-1) \cdot \phi\left(\frac{1}{(c_1 + c_2)(n-1)} \sum_{j \in [n] \setminus \{i\}} (c_1 \boldsymbol{v}_j^\top \boldsymbol{u}_i + c_2 \boldsymbol{u}_j^\top \boldsymbol{u}_i - (c_1 + c_2)\boldsymbol{v}_i^\top \boldsymbol{u}_i)\right)\right)
$$

$$
+ \frac{1}{2n} \sum_{i \in [n]} \psi\left((c_1 + c_2)(n-1) \cdot \phi\left(\frac{1}{(c_1 + c_2)(n-1)} \sum_{j \in [n] \setminus \{i\}} (c_1 \boldsymbol{u}_j^\top \boldsymbol{v}_i + c_2 \boldsymbol{v}_j^\top \boldsymbol{v}_i - (c_1 + c_2)\boldsymbol{u}_i^\top \boldsymbol{v}_i)\right)\right)
$$

$$
\geq \psi\left((c_1 + c_2)(n-1) \cdot \phi\left(\frac{1}{2n} \sum_{i \in [n]} \cdot \frac{1}{(c_1 + c_2)(n-1)} \sum_{j \in [n] \setminus \{i\}} (c_1 \boldsymbol{v}_j^\top \boldsymbol{u}_i + c_2 \boldsymbol{u}_j^\top \boldsymbol{u}_i - (c_1 + c_2)\boldsymbol{v}_i^\top \boldsymbol{u}_i)\right.\right.
$$

$$
\left.\left. + \frac{1}{2n} \sum_{i \in [n]} \cdot \frac{1}{(c_1 + c_2)(n-1)} \sum_{j \in [n] \setminus \{i\}} (c_1 \boldsymbol{u}_j^\top \boldsymbol{v}_i + c_2 \boldsymbol{v}_j^\top \boldsymbol{v}_i - (c_1 + c_2)\boldsymbol{u}_i^\top \boldsymbol{v}_i)\right)\right)
\tag{25}
$$

$$
= h_{(c_1,c_2)}\left(\frac{1}{2(c_1 + c_2)n(n-1)} \sum_{i \in [n]} \sum_{j \in [n] \setminus \{i\}} (c_1 2\boldsymbol{u}_j^\top \boldsymbol{v}_i + c_2(\boldsymbol{u}_j^\top \boldsymbol{u}_i + \boldsymbol{v}_j^\top \boldsymbol{v}_i) - 2(c_1 + c_2)\boldsymbol{u}_i^\top \boldsymbol{v}_i)\right)
$$

where the inequality in (25) holds for Jensen's inequality. Equality in (25) holds if the arguments of $h_{(c_1,c_2)}$ are constant for all $i \in [n]$. Moreover, from Jensen's inequality, we have

$$
\mathbb{E}_{(\boldsymbol{U},\boldsymbol{V}) \sim f_\sharp \hat{p}_{\text{pos}}^{[n]}} [\mathcal{L}_{\text{info-sym}}(\boldsymbol{U}, \boldsymbol{V})]
$$

$$
\geq \mathbb{E}_{(\boldsymbol{U},\boldsymbol{V}) \sim f_\sharp \hat{p}_{\text{pos}}^{[n]}} \left[ h_{(c_1,c_2)}\left(\frac{1}{2(c_1 + c_2)n(n-1)} \sum_{i \in [n]} \sum_{j \in [n] \setminus \{i\}} (c_1 2\boldsymbol{u}_j^\top \boldsymbol{v}_i + c_2(\boldsymbol{u}_j^\top \boldsymbol{u}_i + \boldsymbol{v}_j^\top \boldsymbol{v}_i) - 2(c_1 + c_2)\boldsymbol{u}_i^\top \boldsymbol{v}_i)\right) \right]
$$

$$
\geq h_{(c_1,c_2)}\left(\frac{1}{2(c_1 + c_2)} \mathbb{E}_{(\boldsymbol{U},\boldsymbol{V}) \sim f_\sharp \hat{p}_{\text{pos}}^{[n]}} \left[\frac{1}{n(n-1)} \sum_{i \in [n]} \sum_{j \in [n] \setminus \{i\}} (c_1 2\boldsymbol{u}_j^\top \boldsymbol{v}_i + c_2(\boldsymbol{u}_j^\top \boldsymbol{u}_i + \boldsymbol{v}_j^\top \boldsymbol{v}_i) - 2(c_1 + c_2)\boldsymbol{u}_i^\top \boldsymbol{v}_i)\right]\right).
$$

Therefore, we only have to minimize

$$
\mathbb{E}_{(\boldsymbol{U},\boldsymbol{V}) \sim f_\sharp \hat{p}_{\text{pos}}^{[n]}} \left[\frac{1}{n(n-1)} \sum_{i \in [n]} \sum_{j \in [n] \setminus \{i\}} (c_1 2\boldsymbol{u}_j^\top \boldsymbol{v}_i + c_2(\boldsymbol{u}_j^\top \boldsymbol{u}_i + \boldsymbol{v}_j^\top \boldsymbol{v}_i) - 2(c_1 + c_2)\boldsymbol{u}_i^\top \boldsymbol{v}_i)\right].
$$

Now, consider each case of $(c_1, c_2) \in \{(0, 1), (1, 0), (1, 1)\}$.

For the case of $(c_1, c_2) = (1, 1)$, by using Lemma C.7, we have

$$
\mathbb{E}_{(\boldsymbol{U},\boldsymbol{V}) \sim f_\sharp \hat{p}_{\text{pos}}^{[n]}} \left[\frac{1}{n(n-1)} \sum_{i \in [n]} \sum_{j \in [n] \setminus \{i\}} (2\boldsymbol{u}_j^\top \boldsymbol{v}_i + \boldsymbol{u}_j^\top \boldsymbol{u}_i + \boldsymbol{v}_j^\top \boldsymbol{v}_i - 4\boldsymbol{u}_i^\top \boldsymbol{v}_i)\right]
$$

$$
\geq \mathbb{E}_{(\boldsymbol{U},\boldsymbol{V}) \sim f_\sharp \hat{p}_{\text{pos}}^{[n]}} \left[-\frac{2}{n(n-1)} \sum_{i \in [n]} \boldsymbol{u}_i^\top \boldsymbol{v}_i - \frac{2}{n-1} - \frac{4}{n} \sum_{i \in [n]} \boldsymbol{u}_i^\top \boldsymbol{v}_i\right].
$$

For the case of $(c_1, c_2) = (0, 1)$, by using Lemma C.8, we have

$$
\mathbb{E}_{(\boldsymbol{U},\boldsymbol{V}) \sim f_\sharp \hat{p}_{\text{pos}}^{[n]}} \left[\frac{1}{n(n-1)} \sum_{i \in [n]} \sum_{j \in [n] \setminus \{i\}} (\boldsymbol{u}_j^\top \boldsymbol{u}_i + \boldsymbol{v}_j^\top \boldsymbol{v}_i - 2\boldsymbol{u}_i^\top \boldsymbol{v}_i)\right] \geq \mathbb{E}_{(\boldsymbol{U},\boldsymbol{V}) \sim f_\sharp \hat{p}_{\text{pos}}^{[n]}} \left[-\frac{2}{n-1} - \frac{2}{n} \sum_{i \in [n]} \boldsymbol{u}_i^\top \boldsymbol{v}_i\right].
$$

For the case of $(c_1, c_2) = (1, 0)$, by using Lemma C.6, we have

$$\mathbb{E}_{(U,V) \sim f_\sharp \hat{p}_{\text{pos}}^{[n]}} \left[ \frac{1}{n(n-1)} \sum_{i \in [n]} \sum_{j \in [n] \setminus \{i\}} \left( 2 u_j^\top v_i - 2 u_i^\top v_i \right) \right]$$

$$\geq \mathbb{E}_{(U,V) \sim f_\sharp \hat{p}_{\text{pos}}^{[n]}} \left[ \frac{n-2}{2n(n-1)} \sum_{i \in [n]} u_i^\top v_i - \frac{n}{2(n-1)} - \frac{2}{n(n-1)} \sum_{i \in [n]} \sum_{j \in [n] \setminus \{i\}} u_i^\top v_i \right]$$

$$= -\frac{2}{n(n-1)} + \frac{-n^3 + n^2 + n - 2}{2n(n-1)} \cdot \mathbb{E}_{(U,V) \sim f_\sharp \hat{p}_{\text{pos}}^{[n]}} \left[ \sum_{i \in [n]} u_i^\top v_i \right].$$

Therefore, for every cases of $(c_1, c_2)$, $u_i^\top v_i = 1$ holds for all positive pairs $(u_i, v_i) \sim f_\sharp^\star \hat{p}_{\text{pos}}^i$ and $i \in [n]$. To achieve the all equality conditions, $u^\top v = -\frac{1}{n-1}$ must hold for all negative pairs $(u, v) \sim f_\sharp^\star \hat{p}_{\text{neg}}$. Therefore, embedding similarities for the full-batch optimal encoder $f^\star$ in (1) satisfy

$$s(f^\star; \hat{p}_{\text{pos}}) = 1, \qquad s(f^\star; \hat{p}_{\text{neg}}) = -\frac{1}{n-1}.$$

Second, consider the case of $\mathcal{L}_{\text{ind-add}}(U, V)$ in Def. 3.2. From Jensen's inequality, we have

$$\mathcal{L}_{\text{ind-add}}(U, V) = -\frac{1}{n} \sum_{i \in [n]} \phi(u_i^\top v_i) + \frac{c_1}{n(n-1)} \sum_{i \neq j \in [n]} \psi(u_i^\top v_j) + \frac{c_2}{2n(n-1)} \sum_{i \neq j \in [n]} \left( \psi(u_i^\top u_j) + \psi(v_i^\top v_j) \right)$$

$$\geq -\phi\left( \frac{1}{n} \sum_{i \in [n]} u_i^\top v_i \right) + \frac{c_1}{n(n-1)} \sum_{i \neq j \in [n]} \psi(u_i^\top v_j) + \frac{c_2}{2n(n-1)} \sum_{i \neq j \in [n]} \left( \psi(u_i^\top u_j) + \psi(v_i^\top v_j) \right)$$

$$\geq -\phi\left( \frac{1}{n} \sum_{i \in [n]} u_i^\top v_i \right) + \psi\left( \frac{1}{(2c_1 + c_2)n(n-1)} \sum_{i \neq j \in [n]} \left( 2c_1 u_i^\top v_j + c_2 u_i^\top u_j + c_2 v_i^\top v_j \right) \right).$$

Equality conditions for both inequality is $\phi$ and $\psi$ are applied to a constant argument.

Now, consider each case of $(c_1, c_2) \in \{(0, 1), (1, 0), (1, 1)\}$.

For the case of $(c_1, c_2) = (1, 1)$, by using Lemma C.7, we have

$$\mathcal{L}_{\text{ind-add}}(U, V) \geq -\phi\left( \frac{1}{n} \sum_{i \in [n]} u_i^\top v_i \right) + \psi\left( \frac{1}{3n(n-1)} \sum_{i \neq j \in [n]} \left( 2 u_i^\top v_j + u_i^\top u_j + v_i^\top v_j \right) \right)$$

$$\geq -\phi\left( \frac{1}{n} \sum_{i \in [n]} u_i^\top v_i \right) + \psi\left( -\frac{2}{3n(n-1)} \sum_{i \in [n]} u_i^\top v_i - \frac{2}{3(n-1)} \right).$$

Note that $\phi$ and $\psi$ are increasing functions. Therefore, by using the similar manner in the proof of $\mathcal{L}_{\text{info-sym}}(U, V)$ case above, embedding similarities in Def. 4.1 of the full-batch optimal encoder $f^\star$ in (1) satisfy

$$s(f^\star; \hat{p}_{\text{pos}}) = 1, \qquad s(f^\star; \hat{p}_{\text{neg}}) = -\frac{1}{n-1}.$$

For the case of $(c_1, c_2) = (0, 1)$, by using Lemma C.8, we have

$$\mathcal{L}_{\text{ind-add}}(U, V) \geq -\phi\left( \frac{1}{n} \sum_{i \in [n]} u_i^\top v_i \right) + \psi\left( \frac{1}{n(n-1)} \sum_{i \neq j \in [n]} \left( u_i^\top u_j + v_i^\top v_j \right) \right)$$

$$\geq -\phi\left( \frac{1}{n} \sum_{i \in [n]} u_i^\top v_i \right) + \psi\left( -\frac{2}{n-1} \right)$$

Note that $\phi$ and $\psi$ are increasing functions. Therefore, by using the similar manner in the case of $\mathcal{L}_{\text{info-sym}}(U, V)$ in Def. 3.1, embedding similarities in Def. 4.1 of the full-batch optimal encoder $f^\star$ in (1) satisfy

$$s(f^\star; \hat{p}_{\text{pos}}) = 1, \qquad s(f^\star; \hat{p}_{\text{neg}}) = -\frac{1}{n-1}.$$

For the case of $(c_1, c_2) = (1, 0)$, by using Lemma C.6, we have

$$
\begin{aligned}
\mathcal{L}_{\text{ind-add}}(U, V) &\geq -\phi\left(\frac{1}{n}\sum_{i\in[n]} u_i^\top v_i\right) + \psi\left(\frac{1}{n(n-1)}\sum_{i\neq j\in[n]} u_i^\top v_j\right) \\
&\geq -\phi\left(\frac{1}{n}\sum_{i\in[n]} u_i^\top v_i\right) + \psi\left(\frac{n-2}{2n(n-1)}\sum_{i=1}^{n} u_i^\top v_i - \frac{n}{2(n-1)}\right) \\
&= h\left(\frac{1}{n}\sum_{i\in[n]} u_i^\top v_i\right)
\end{aligned}
$$

where the function $h(\cdot)$ is defined as

$$h(x) := -\phi(x) + \psi\left(\frac{n-2}{2(n-1)}\cdot x - \frac{n}{2(n-1)}\right).$$

Note that both $-\phi(\cdot)$ and $\psi(\cdot)$ are differentiable and convex functions, therefore $h(\cdot)$ is also a differentiable and convex function. Therefore, to attain the minimum at $x = 1$, $h'(1) < 0$ holds. Then,

$$0 > h'(1) = -\phi'(1) + \frac{n-2}{2(n-1)}\cdot\psi'\left(\frac{n-2}{2(n-1)} - \frac{n}{2(n-1)}\right) = -\phi'(1) + \frac{n-2}{2(n-1)}\cdot\psi'\left(-\frac{1}{n-1}\right),$$

which is equal to

$$\phi'(1) > \frac{n-2}{2(n-1)}\cdot\psi'\left(-\frac{1}{n-1}\right).$$

Therefore, by using the similar manner in the case of $\mathcal{L}_{\text{info-sym}}(U, V)$ in Def. 3.1, embedding similarities in Def. 4.1 of the full-batch optimal encoder $f^\star$ in (1) satisfy

$$s(f^\star; \hat{p}_{\text{pos}}) = 1, \qquad s(f^\star; \hat{p}_{\text{neg}}) = -\frac{1}{n-1}.$$

if $\phi'(1) > \frac{n-2}{2(n-1)}\cdot\psi'\left(-\frac{1}{n-1}\right)$. On the other hand, if $\phi'(1) < \frac{n-2}{2(n-1)}\cdot\psi'\left(-\frac{1}{n-1}\right)$, embedding similarities in Def. 4.1 of the full-batch optimal encoder $f^\star$ in (1) satisfy

$$s(f^\star; \hat{p}_{\text{pos}}) < 1, \qquad s(f^\star; \hat{p}_{\text{neg}}) < -\frac{1}{n-1}.$$

Moreover, the existence of the embedding for $d \geq n$ can be shown in Proposition 1 in Lee et al. (2024) $\qquad\square$

**Example C.11.** *Consider the sigmoid contrastive loss $\mathcal{L}_{\text{sig}}(U, V)$ (Zhai et al., 2023), defined as*

$$\mathcal{L}_{\text{sig}}(U, V) := \frac{1}{n}\sum_{i\in[n]}\log\left(1 + \exp\left(-tu_i^\top v_i\right)\cdot\exp(b)\right) + \frac{1}{n}\sum_{i\neq j\in[n]}\log\left(1 + \exp\left(tu_i^\top v_j\right)\cdot\exp(-b)\right),$$

*where $t > 0$ and $b \in \mathbb{R}$ are hyperparameters. This loss follows the form of $\mathcal{L}_{\text{ind-add}}(U, V)$ in Def. 3.2, where $(c_1, c_2) = (1, 0)$, $\phi(x) = -\log(1 + \exp(-tx + b))$, and $\psi(x) = (n-1)\cdot\log(1 + \exp(tx - b))$.*

*If hyperparameters $t$ and $b$ are chosen such that*

$$\frac{1 + \exp\left(\frac{t}{n-1} + b\right)}{1 + \exp(t - b)} < \frac{n-2}{2},$$

*embedding similarities of the full-batch optimal encoder $f^\star$ in (1) satisfy*

$$\boldsymbol{s}(f^\star; \hat{p}_{\text{pos}}) < 1, \qquad \boldsymbol{s}(f^\star; \hat{p}_{\text{neg}}) < -\frac{1}{n-1}.$$

*Proof.* The sigmoid contrastive loss $\mathcal{L}_{\text{sig}}(\boldsymbol{U}, \boldsymbol{V})$ follows the loss form in Def. 3.2, see Appendix A.2. Consequently, by Theorem C.10, it suffices to verify the condition of

$$\phi'(1) < \frac{n-2}{2(n-1)} \cdot \psi'\left(-\frac{1}{n-1}\right), \tag{26}$$

where $\phi(x) = -\log(1 + \exp(-tx + b))$ and $\psi(x) = (n-1)\log(1 + \exp(tx - b))$. Taking the derivative of $\phi(x)$, we obtain

$$\phi'(x) = \frac{t \exp(-tx + b)}{1 + \exp(-tx + b)} = \frac{t}{1 + \exp(tx - b)},$$

and differentiating $\psi(x)$ yields

$$\psi'(x) = (n-1) \cdot \frac{t \exp(tx - b)}{1 + \exp(tx - b)}.$$

By plugging the derivative values into (26), we get

$$\frac{t}{1 + \exp(t - b)} < \frac{n-2}{2(n-1)} \cdot \frac{(n-1)t \exp\left(-\frac{t}{n-1} - b\right)}{1 + \exp\left(-\frac{t}{n-1} - b\right)},$$

which simplifies to

$$\frac{1}{1 + \exp(t - b)} < \frac{n-2}{2(n-1)} \cdot \frac{(n-1)\exp\left(-\frac{t}{n-1} - b\right)}{1 + \exp\left(-\frac{t}{n-1} - b\right)} = \frac{n-2}{2} \cdot \frac{1}{\exp\left(\frac{t}{n-1} + b\right) + 1}.$$

Therefore, if hyperparameters $t$ and $b$ satisfy

$$\frac{1 + \exp\left(\frac{t}{n-1} + b\right)}{1 + \exp(t - b)} < \frac{n-2}{2}, \tag{27}$$

following from Theorem C.10, the similarities of the full-batch optimal encoder $f^\star$ in (1) satisfy

$$\boldsymbol{s}(f^\star; \hat{p}_{\text{pos}}) < 1, \qquad \boldsymbol{s}(f^\star; \hat{p}_{\text{neg}}) < -\frac{1}{n-1}.$$

$\square$

## C.4. Proofs for Mini-Batch CL

**Definition C.12** (Simplex ETF). A set of $n$ vectors $\boldsymbol{U}$ on the $d$-dimensional unit sphere is called $(n-1)$-simplex ETF, if

$$\|\boldsymbol{u}\|_2^2 = 1 \text{ and } \boldsymbol{u}^\top \boldsymbol{v} = -\frac{1}{n-1}, \qquad\qquad \forall \boldsymbol{u} \neq \boldsymbol{v} \in \boldsymbol{U}.$$

Note that $(n-1)$-simplex ETF exists when $d \geq n - 1$.

**Lemma C.13.** *Let a set of $n$ vectors $U$ be a $(n-1)$-simplex ETF on the $d$-dimensional unit sphere with $d \geq n - 1$. Then, the following holds:*

$$\sum_{u \in U} u = 0$$

*Proof.* By the definition of a $(n-1)$-simplex ETF, each vector $u \in U$ satisfies $\|u\|^2 = 1$ and the pairwise inner product for any $u, v \in U$ with $u \neq v$ is $u^\top v = -\frac{1}{n-1}$. Then,

$$\left\| \sum_{u \in U} u \right\|_2^2 = \left( \sum_{u \in U} u \right)^\top \left( \sum_{u \in U} u \right) = \sum_{u \in U} u^\top u + \sum_{u \neq v \in U} u^\top v = n \cdot 1 + n(n-1) \cdot \left( -\frac{1}{n-1} \right) = 0.$$

Since the squared norm is zero, we conclude $\sum_{u \in U} u = 0$. $\square$

**Lemma C.14.** *Let a set of $n$ vectors $U$ be a $(n-1)$-simplex ETF, and a set of $m$ vectors $V$ be a $(m-1)$-simplex ETF, where all vectors are on the $d$-dimensional unit sphere with $d \geq \max(n, m) - 1$. Then, the following holds:*

$$\frac{1}{|U \cup V|(|U \cup V| - 1)} \sum_{u \neq v \in U \cup V} u^\top v = -\frac{1}{n + m - 1}.$$

*Here, $U \cup V$ denotes the concatenation of the two sets, representing the full collection of all $n + m$ vectors from $U$ and $V$.*

*Proof.* From Def C.12, we have

$$\frac{1}{n(n-1)} \sum_{u \neq v \in U} u^\top v = -\frac{1}{n-1}, \qquad \frac{1}{m(m-1)} \sum_{u \neq v \in V} u^\top v = -\frac{1}{m-1}.$$

Moreover, Lemma C.13 implies that

$$\sum_{u \in U} \sum_{v \in V} u^\top v = \left( \sum_{u \in U} u \right)^\top \left( \sum_{v \in V} v \right) = 0^\top 0 = 0.$$

The total sum of pairwise inner products for the combined set $U \cup V$ can be written as

$$\begin{aligned}
\sum_{u \neq v \in U \cup V} u^\top v &= \sum_{u \in U \cup V} \sum_{v \in U \cup V \setminus \{u\}} u^\top v \\
&= \sum_{u \in U} \sum_{v \in U \cup V \setminus \{u\}} u^\top v + \sum_{u \in V} \sum_{v \in U \cup V \setminus \{u\}} u^\top v \\
&= \sum_{u \in U} \sum_{v \in U \setminus \{u\}} u^\top v + \sum_{u \in U} \sum_{v \in V} u^\top v + \sum_{u \in V} \sum_{v \in U} u^\top v + \sum_{u \in V} \sum_{v \in V \setminus \{u\}} u^\top v \\
&= \sum_{u \neq v \in U} u^\top v + \sum_{u \in U, v \in V} u^\top v + \sum_{v \in V, u \in U} u^\top v + \sum_{u \neq v \in V} u^\top v \\
&= \sum_{u \neq v \in U} u^\top v + \sum_{u \neq v \in V} u^\top v \qquad (28) \\
&= n(n-1) \cdot \left( -\frac{1}{n-1} \right) + m(m-1) \cdot \left( -\frac{1}{m-1} \right) \\
&= -n - m,
\end{aligned}$$

where the equality in (28) holds because the total pairwise inner product sums within $U$ and within $V$ are zero.

Therefore, we obtain

$$\frac{1}{|U \cup V|(|U \cup V| - 1)} \sum_{u \neq v \in U \cup V} u^\top v = \frac{1}{(n+m)(n+m-1)}(-n-m) = -\frac{1}{n+m-1}.$$

$\square$

**Lemma C.15.** *Suppose $d \geq n - 1$. Let $V$ and $W$ be $d \times n$ matrices whose columns form $(n-1)$-simplex ETF on the $d$-dimensional unit sphere. Then, there exist an orthogonal matrix $P \in \mathbb{R}^{d \times d}$ such that $V = PW$.*

*Proof.* From the definition of simplex ETF in Def. C.12, the Gram matrices satisfy

$$V^\top V = W^\top W,$$

where each diagonal entry is 1 and each off-diagonal entry is $-\frac{1}{n-1}$. Then, from Theorem 7.3.11 in Horn & Johnson (2012) there exist an orthogonal matrix $P \in \mathbb{R}^{d \times d}$ such that $V = PW$. □

**Theorem C.16.** *Suppose $d \geq m - 1$. Let the contrastive loss $\mathcal{L}(U, V)$ be one of the forms in Theorem 5.1. Define $f_{\text{batch}}^\star$ as the optimal encoder that minimizes the fixed mini-batch loss, given by*

$$f_{\text{batch}}^\star := \arg\min_f \mathbb{E}_{(U,V) \sim f_\sharp \hat{p}_{\text{pos}}^{[n]}} \left[ \sum_{k \in [b]} \mathcal{L}(U_{I_k}, V_{I_k}) \right],$$

*where $I_k := [m(k-1) + 1 : mk]$ for $k \in [b]$.*

*Then, embedding similarities for the mini-batch optimal encoder $f_{\text{batch}}^\star$ satisfy*

$$s(f_{\text{batch}}^\star; \hat{p}_{\text{pos}}) = 1,$$
$$\mathbb{E}[s(f_{\text{batch}}^\star; \hat{p}_{\text{neg}})] = -\frac{1}{n-1},$$
$$\text{Var}[s(f_{\text{batch}}^\star; \hat{p}_{\text{neg}})] \in \left[ \frac{n-m}{(m-1)(n-1)^2}, \frac{n(n-m)}{(m-1)(n-1)^2} \right]. \tag{29}$$

*A necessary condition for attaining the minimum variance of negative-pair similarities in (11) is $d \geq b(m-1)$.*

*Proof.* Note that

$$\mathbb{E}_{(U,V) \sim f_\sharp \hat{p}_{\text{pos}}^{[n]}} \left[ \sum_{k \in [b]} \mathcal{L}(U_{I_k}, V_{I_k}) \right] = \sum_{k \in [b]} \mathbb{E}_{(U,V) \sim f_\sharp \hat{p}_{\text{pos}}^{I_k}} [\mathcal{L}(U_{I_k}, V_{I_k})],$$

by applying Theorem 5.1 to each batch, $m$ random vectors in each batch are degenerated to construct the $(m-1)$-simplex ETF in Def. C.12. Therefore, for $k \in [b]$, we have:

$$u^\top v = 1 \qquad\qquad\qquad \forall(u, v) \sim f_\sharp \hat{p}_{\text{pos}}^{I_k},$$
$$u^\top v = -\frac{1}{m-1} \qquad\qquad \forall(u, v) \sim f_\sharp \hat{p}_{\text{neg}}^{I_k}. \tag{30}$$

In what follows, for all positive pairs, we have

$$u^\top v = 1 \qquad \forall(u, v) \sim f_\sharp^\star \hat{p}_{\text{pos}},$$

which is equal to

$$s(f_{\text{batch}}^\star; \hat{p}_{\text{pos}}) = 1.$$

For $k \in [b]$, let $U^{(k)}$ and $V^{(k)}$ denote $d \times m$ random matrices, where the columns represent the vectors in the $k$-th batch. Additionally, define $U$ and $V$ as $d \times mb$ random matrices formed by concatenation of all corresponding batch matrices, *i.e.,* $U := [U^{(1)}, U^{(2)}, \cdots, U^{(b)}]$ and $V := [V^{(1)}, V^{(2)}, \cdots, V^{(b)}]$.

Let $W$ be a $d \times m$ matrix whose columns form $(m-1)$-simplex ETF in Def. C.12. From Lemma C.15, for $k \in [b]$, there exist orthogonal matrices $P^{(k)} \in \mathbb{R}^{d \times d}$ such that $V^{(k)} = P^{(k)} W$. Moreover, based on the singular value decomposition, let $W = W_1 \Sigma W_2^\top$, where $W_1$ is a $d \times d$ orthogonal matrix, $W_2$ is an $m \times m$ orthogonal matrix, and $\Sigma$ is a $d \times m$ rectangular diagonal matrix with non-negative values of $\sigma_1, \sigma_2, \cdots, \sigma_m$ on the diagonal.

For all $k \in [b]$, we have

$$\left\| \left( U^{(k)} \right)^\top V^{(k)} \right\|_F^2 = \left\| \left( V^{(k)} \right)^\top V^{(k)} \right\|_F^2 = \| W^\top W \|_F^2 = m \cdot 1 + m(m-1) \cdot \left( -\frac{1}{m-1} \right)^2 = \frac{m^2}{m-1}.$$

For all $k_1 \neq k_2 \in [b]$, we have

$$\left\| \left( U^{(k_1)} \right)^\top V^{(k_2)} \right\|_F^2 = \left\| \left( V^{(k_1)} \right)^\top V^{(k_2)} \right\|_F^2 = \left\| W^\top \left( P^{(k_1)} \right)^\top P^{(k_2)} W \right\|_F^2 \geq 0,$$

where the minimum value of zero is achieved if $\left( P^{(k_1)} \right)^\top P^{(k_2)}$ is the $d \times d$ consisting entirely of zero elements. On the other hand,

$$
\begin{aligned}
\left\| \left( U^{(k_1)} \right)^\top V^{(k_2)} \right\|_F^2 &= \left\| \left( V^{(k_1)} \right)^\top V^{(k_2)} \right\|_F^2 \\
&= \left\| W^\top \left( P^{(k_1)} \right)^\top P^{(k_2)} W \right\|_F^2 \\
&= \left\| W_2 \Sigma W_1^\top \left( P^{(k_1)} \right)^\top P^{(k_2)} W_1 \Sigma W_2^\top \right\|_F^2 \\
&= \left\| \Sigma W_1^\top \left( P^{(k_1)} \right)^\top P^{(k_2)} W_1 \Sigma \right\|_F^2 \\
&= \left\| \Sigma P_1^\top P_2 \Sigma \right\|_F^2,
\end{aligned}
$$

where $P_1 := P^{(k_1)} W_1$ and $P_2 := P^{(k_2)} W_1$ are $m \times m$ orthogonal matrices, and each $P_{1i}$ and $P_{2i}$ is a column vector of $P_1$ and $P_2$, respectively. Since $P_1$ and $P_2$ are orthogonal matrices, their columns are orthonormal vectors, respectively. Then,

$$\left\| \left( U^{(k_1)} \right)^\top V^{(k_2)} \right\|_F^2 = \left\| \Sigma P_1^\top P_2 \Sigma \right\|_F^2 = \sum_{i \in [m]} \sigma_i^2 P_{1i}^\top P_{2i} \geq \sum_{i \in [m]} \sigma_i^2 = \| \Sigma \Sigma \|_F^2 = \| W^\top W \|_F^2 = \frac{m^2}{m-1},$$

where the maximum value of $m$ is attained if $P_{1i} = P_{2i}$ for all $i \in [m]$, *i.e.*, $P_1 = P_2$.

From Lemma C.14, the expectation of the negative-pair similarity is

$$\mathbb{E} \left[ s(f_{\text{batch}}^\star; \hat{p}_{\text{neg}}) \right] = \mathbb{E}_{(u,v) \sim f_{\text{batch}}^\star \sharp \hat{p}_{\text{neg}}} \left[ u^\top v \right] = -\frac{1}{n-1}.$$

Note that

$$
\begin{aligned}
\mathbb{E} \left[ s(f_{\text{batch}}^\star; \hat{p}_{\text{neg}})^2 \right] &= \mathbb{E}_{(u,v) \sim f_{\text{batch}}^\star \sharp \hat{p}_{\text{neg}}} \left[ \left( u^\top v \right)^2 \right] \\
&= \frac{1}{n^2 - n} \left( \| U^\top V \|_F^2 - n \right) \\
&= \frac{1}{n^2 - n} \sum_{k_1, k_2 \in [b]} \left\| \left( U^{(k_1)} \right)^\top V^{(k_2)} \right\|_F^2 - \frac{1}{n-1} \\
&= \frac{1}{n^2 - n} \sum_{k_1 \neq k_2 \in [b]} \left\| \left( U^{(k_1)} \right)^\top V^{(k_2)} \right\|_F^2 + \frac{1}{n^2 - n} \sum_{k \in [b]} \left\| \left( U^{(k)} \right)^\top V^{(k)} \right\|_F^2 - \frac{1}{n-1}
\end{aligned}
$$

$$= \frac{1}{n^2 - n} \sum_{k_1 \neq k_2 \in [b]} \left\| \left( U^{(k_1)} \right)^\top V^{(k_2)} \right\|_F^2 + \frac{1}{n^2 - n} \cdot b \cdot \frac{m^2}{m-1} - \frac{1}{n-1}$$

$$= \frac{1}{n^2 - n} \sum_{k_1 \neq k_2 \in [b]} \left\| \left( U^{(k_1)} \right)^\top V^{(k_2)} \right\|_F^2 + \frac{1}{(m-1)(n-1)}$$

$$\in \left[ 0 + \frac{1}{(m-1)(n-1)}, \frac{1}{n^2 - n} \cdot b(b-1) \cdot \frac{m^2}{m-1} + \frac{1}{(m-1)(n-1)} \right],$$

where the range is equal to $\left[ \frac{1}{(m-1)(n-1)}, \frac{m(b-1)+1}{(m-1)(n-1)} \right]$.

Therefore, the variance of the negative-pair similarity is

$$\mathrm{Var}\left[ s(f^\star_{\mathrm{batch}}; \hat{p}_{\mathrm{neg}}) \right] = \mathbb{E}\left[ s(f^\star_{\mathrm{batch}}; \hat{p}_{\mathrm{neg}})^2 \right] - \mathbb{E}\left[ s(f^\star_{\mathrm{batch}}; \hat{p}_{\mathrm{neg}}) \right]^2$$

$$= \mathbb{E}\left[ s(f^\star_{\mathrm{batch}}; \hat{p}_{\mathrm{neg}})^2 \right] - \frac{1}{(n-1)^2}$$

$$\in \left[ \frac{n-m}{(m-1)(n-1)^2}, \frac{n(n-m)}{(m-1)(n-1)^2} \right].$$

□

**Lemma C.17.** *Let $m_1$ and $m_2$ be natural numbers with $m_1 \leq m_2$. For any positive values $\{c_i > 0\}_{i \in [m_2]}$, the following inequality holds:*

$$\frac{m_1 \sum_{i \in [m_1]} c_i}{m_2 \sum_{i \in [m_2]} c_i} \leq 1,$$

*where the equality condition is $m_1 = m_2$.*

*Proof.* Since $m_2 > 0$ and $m_2 - m_1 \geq 0$, we have

$$m_2 \sum_{i \in [m_2]} c_i - m_1 \sum_{i \in [m_1]} c_i = m_2 \sum_{i \in [m_2] \setminus [m_1]} c_i + (m_2 - m_1) \sum_{i \in [m_1]} c_i \geq 0,$$

where the equality condition is $m_1 = m_2$. Note that $m_1 \sum_{i \in [m_1]} c_i > 0$. Rearranging the above yields

$$\frac{m_1 \sum_{i \in [m_1]} c_i}{m_2 \sum_{i \in [m_2]} c_i} \leq 1.$$

□

**Theorem C.18.** *Consider the InfoNCE loss $\mathcal{L}_{\mathrm{InfoNCE}}(U, V)$ (Oord et al., 2018), which corresponds to the loss $\mathcal{L}_{\mathrm{info\text{-}sym}}(U, V)$ in Def. 3.1 where $\phi(x) = \exp(x/t)$ for some $t > 0$, $\psi(x) = \log(1 + x)$, and $(c_1, c_2) = (1, 0)$.*

*For any two integers $m_1, m_2 \in [n]$ such that $m_1 \leq m_2$, the gradient of the InfoNCE loss with respect to a negative-pair similarity satisfies the following inequalities for any distinct indices $i \neq j \in [m]$.*

$$\mathbb{E}_{(U,V) \sim f_\sharp \hat{p}^{1:m_1}} \left[ -\frac{\partial}{\partial \left( u_i^\top v_j \right)} \mathcal{L}_{\mathrm{InfoNCE}}(U, V) \right] \leq \mathbb{E}_{(U,V) \sim f_\sharp \hat{p}^{1:m_2}} \left[ -\frac{\partial}{\partial \left( u_i^\top v_j \right)} \mathcal{L}_{\mathrm{InfoNCE}}(U, V) \right] \leq 0.$$

*Moreover, the equality condition of the first inequality is $m_1 = m_2$.*

*Proof.* Without loss of generality, suppose there are $m$ positive pairs in $(U, V)$. Then, the InfoNCE loss is given as follows.

$$\mathcal{L}(U, V) = \frac{1}{m} \sum_{i \in [m]} \log \left( 1 + \sum_{j \in [m] \setminus \{i\}} \exp(u_i^\top (v_j - v_i)/t) \right) + \frac{1}{m} \sum_{i \in [m]} \log \left( 1 + \sum_{j \in [m] \setminus \{i\}} \exp((u_j - u_i)^\top v_i/t) \right).$$

Following the gradients analysis in Wang & Liu (2021), the partial derivatives of the loss with respect to negative pair are derived as follows. In particular, for all $i \neq j \in [m]$, we have

$$
\frac{\partial}{\partial \left( \boldsymbol{u}_i^\top \boldsymbol{v}_j \right)} \mathcal{L}(\boldsymbol{U}, \boldsymbol{V}) = \frac{1}{m} \cdot \frac{\partial}{\partial \left( \boldsymbol{u}_i^\top \boldsymbol{v}_j \right)} \log \left( 1 + \sum_{j' \in [m] \setminus \{i\}} \exp(\boldsymbol{u}_i^\top (\boldsymbol{v}_{j'} - \boldsymbol{v}_i)/t) \right)
$$

$$
+ \frac{1}{m} \cdot \frac{\partial}{\partial \left( \boldsymbol{u}_i^\top \boldsymbol{v}_j \right)} \log \left( 1 + \sum_{i' \in [m] \setminus \{i\}} \exp((\boldsymbol{u}_{i'} - \boldsymbol{u}_j)^\top \boldsymbol{v}_j/t) \right)
$$

$$
= \frac{1}{m} \cdot \frac{\exp(\boldsymbol{u}_i^\top (\boldsymbol{v}_j - \boldsymbol{v}_i)/t)/t}{1 + \sum_{j' \in [m] \setminus \{i\}} \exp(\boldsymbol{u}_i^\top (\boldsymbol{v}_{j'} - \boldsymbol{v}_i)/t)} + \frac{1}{m} \cdot \frac{\exp((\boldsymbol{u}_i - \boldsymbol{u}_j)^\top \boldsymbol{v}_j/t)/t}{1 + \sum_{i' \in [m] \setminus \{i\}} \exp((\boldsymbol{u}_{i'} - \boldsymbol{u}_j)^\top \boldsymbol{v}_j/t)}
$$

$$
= \frac{1}{mt} \cdot \frac{\exp(\boldsymbol{u}_i^\top \boldsymbol{v}_j/t)}{\sum_{j' \in [m]} \exp(\boldsymbol{u}_i^\top \boldsymbol{v}_{j'}/t)} + \frac{1}{mt} \cdot \frac{\exp(\boldsymbol{u}_i^\top \boldsymbol{v}_j/t)}{\sum_{i' \in [m]} \exp(\boldsymbol{u}_{i'}^\top \boldsymbol{v}_i/t)}
$$

$$
= \frac{1}{mt} \left( \frac{\exp(\boldsymbol{u}_i^\top \boldsymbol{v}_j/t)}{\sum_{j \in [m]} \exp(\boldsymbol{u}_i^\top \boldsymbol{v}_j/t)} + \frac{\exp(\boldsymbol{u}_i^\top \boldsymbol{v}_j/t)}{\sum_{i' \in [m]} \exp(\boldsymbol{u}_{i'}^\top \boldsymbol{v}_i/t)} \right)
$$

$$
\geq 0.
$$

Then, for any $m_1 \leq m_2 \leq n$ and $i \neq j \in [m_1]$, the following inequality holds:

$$
\mathbb{E}_{(\boldsymbol{U}, \boldsymbol{V}) \sim f_\sharp \hat{p}_{\text{pos}}^{m_2}} \left[ \frac{\partial}{\partial \left( \boldsymbol{u}_i^\top \boldsymbol{v}_j \right)} \mathcal{L}(\boldsymbol{U}, \boldsymbol{V}) \right] = \mathbb{E}_{(\boldsymbol{U}, \boldsymbol{V}) \sim f_\sharp \hat{p}_{\text{pos}}^{m_2}} \left[ \frac{1}{m_2 t} \left( \frac{\exp(\boldsymbol{u}_i^\top \boldsymbol{v}_j/t)}{\sum_{j \in [m_2]} \exp(\boldsymbol{u}_i^\top \boldsymbol{v}_j/t)} + \frac{\exp(\boldsymbol{u}_i^\top \boldsymbol{v}_j/t)}{\sum_{i' \in [m_2]} \exp(\boldsymbol{u}_{i'}^\top \boldsymbol{v}_i/t)} \right) \right]
$$

$$
= \frac{1}{m_1 t} \mathbb{E}_{(\boldsymbol{U}, \boldsymbol{V}) \sim f_\sharp \hat{p}_{\text{pos}}^{m_2}} \left[ \frac{\exp(\boldsymbol{u}_i^\top \boldsymbol{v}_j/t)}{\sum_{j \in [m_1]} \exp(\boldsymbol{u}_i^\top \boldsymbol{v}_j/t)} \cdot \frac{m_1 \sum_{j \in [m_1]} \exp(\boldsymbol{u}_i^\top \boldsymbol{v}_j/t)}{m_2 \sum_{j \in [m_2]} \exp(\boldsymbol{u}_i^\top \boldsymbol{v}_j/t)} \right]
$$

$$
+ \frac{1}{m_1 t} \mathbb{E}_{(\boldsymbol{U}, \boldsymbol{V}) \sim f_\sharp \hat{p}_{\text{pos}}^{m_2}} \left[ \frac{\exp(\boldsymbol{u}_i^\top \boldsymbol{v}_j/t)}{\sum_{i' \in [m_2]} \exp(\boldsymbol{u}_{i'}^\top \boldsymbol{v}_i/t)} \cdot \frac{m_1 \sum_{i' \in [m_1]} \exp(\boldsymbol{u}_{i'}^\top \boldsymbol{v}_i/t)}{m_2 \sum_{i' \in [m_2]} \exp(\boldsymbol{u}_{i'}^\top \boldsymbol{v}_i/t)} \right]
$$

$$
\leq \frac{1}{m_1 t} \mathbb{E}_{(\boldsymbol{U}, \boldsymbol{V}) \sim f_\sharp \hat{p}_{\text{pos}}^{m_2}} \left[ \frac{\exp(\boldsymbol{u}_i^\top \boldsymbol{v}_j/t)}{\sum_{j \in [m_1]} \exp(\boldsymbol{u}_i^\top \boldsymbol{v}_j/t)} \cdot 1 \right]
$$

$$
+ \frac{1}{m_1 t} \mathbb{E}_{(\boldsymbol{U}, \boldsymbol{V}) \sim f_\sharp \hat{p}_{\text{pos}}^{m_2}} \left[ \frac{\exp(\boldsymbol{u}_i^\top \boldsymbol{v}_j/t)}{\sum_{i' \in [m_2]} \exp(\boldsymbol{u}_{i'}^\top \boldsymbol{v}_i/t)} \cdot 1 \right] \tag{31}
$$

$$
= \frac{1}{m_1 t} \mathbb{E}_{(\boldsymbol{U}, \boldsymbol{V}) \sim f_\sharp \hat{p}_{\text{pos}}^{m_1}} \left[ \frac{\exp(\boldsymbol{u}_i^\top \boldsymbol{v}_j/t)}{\sum_{j \in [m_1]} \exp(\boldsymbol{u}_i^\top \boldsymbol{v}_j/t)} + \frac{\exp(\boldsymbol{u}_i^\top \boldsymbol{v}_j/t)}{\sum_{i' \in [m_1]} \exp(\boldsymbol{u}_{i'}^\top \boldsymbol{v}_i/t)} \right] \tag{32}
$$

$$
= \mathbb{E}_{(\boldsymbol{U}, \boldsymbol{V}) \sim f_\sharp \hat{p}_{\text{pos}}^{m_1}} \left[ \frac{\partial}{\partial \left( \boldsymbol{u}_i^\top \boldsymbol{v}_j \right)} \mathcal{L}(\boldsymbol{U}, \boldsymbol{V}) \right],
$$

where the inequality in (31) follows from Lemma C.17, since the exponential function is strictly positive. The equality condition in (31) is $m_1 = m_2$.

Moreover, equality in (32), where $f_\sharp \hat{p}_{\text{pos}}^{m_2}$ is replaced with $f_\sharp \hat{p}_{\text{pos}}^{m_1}$, holds because the expectation involves only embeddings $(\boldsymbol{U}, \boldsymbol{V})$ with indices in $[m_1]$. This concludes the proof. $\qquad\square$

# D. Experiment Details

In this section, we provide the details of the experiment setup mentioned in Sec. 6. Our implementation is based on the open-source library solo-learn (da Costa et al., 2022) for self-supervised learning. The source code is available at https://github.com/leechungpa/embedding-similarity-cl/.

## D.1. Architecture and Training Details

For all experiments, we use modified ResNet-18 (He et al., 2016; Chen et al., 2020) as the backbone for CIFAR datasets and ResNet-50 for ImageNet-100. For CIFAR datasets, we modify ResNet-18 by replacing the first convolutional layer with a 3×3 kernel at a stride of 1 and removing the initial max pooling step. In contrast, we use the standard ResNet-50 architecture for ImageNet-100 without any modifications. Regardless of the backbone used, we attach a 2-layer MLP as the projection head, which projects representations to a 128-dimensional latent space. Batch normalization is applied to the fully connected layers, with the hidden layer dimension set to 2048 for ImageNet-100 and 512 for the CIFAR datasets.

We follow the data augmentation strategy used in SimCLR (Chen et al., 2020). Specifically, we apply random resized cropping, horizontal flipping, color jittering, and Gaussian blurring. For the CIFAR datasets, the crop size is set to 32, while for ImageNet-100, we use a crop size of 224. These augmentations are applied consistently across all experiments.

For the optimizer, we use stochastic gradient descent (SGD) for 200 epochs. The learning rate is scaled linearly with the batch size as $\text{lr} \times \text{BatchSize}/256$, where the base learning rate is set to 0.3 for the CIFAR datasets and 0.1 for ImageNet-100. A cosine decay schedule is applied, with a weight decay of 0.0001 and SGD momentum set to 0.9. Additionally, we use linear warmup for the first 10 epochs.

We tune the temperature parameter for baseline methods, SimCLR, DCL, and DHEL, by performing a grid search over the range of 0.1 to 0.5 in increments of 0.1 and selecting the temperature value that yielded the best performance for each method. For tuning the proposed loss $\mathcal{L}_{\text{VRNS}}(\mathbf{U}, \mathbf{V})$ in Def. 5.7, we conducted a grid search for $\lambda$ from the set $\{0.1, 0.3, 1, 3, 10, 30, 100\}$.

All experiments were conducted using a single NVIDIA RTX 4090 GPU.

## D.2. Evaluation Details

For the linear evaluation protocol, we remove the projector head and using the pretrained encoder for downstream classification tasks. Specifically, we extract the encoder outputs from the trained model without applying any augmentations. These feature vectors are then normalized and used to train a linear classifier. Following prior works (Kornblith et al., 2019; Lee et al., 2021; Koromilas et al., 2024), we report top 1 accuracy on the downstream dataset. We use SGD, setting the learning rate to 0.1 without weight decay. The classifier is trained for 200 epochs with a batch size of 256.

# E. Additional Experiments on the ImageNet Dataset

Table 4. Effect of our proposed loss (when combined with SimCLR) on the top-1 and top-5 classification accuracies (%). Bold entries indicate the highest accuracy.

| Dataset | Temperature | SimCLR | | SimCLR + Ours | |
|---|---|---|---|---|---|
| | | Top 1 | Top 5 | Top 1 | Top 5 |
| | $t = 0.1$ | 70.14 | 91.14 | 69.50 | 90.80 |
| | $t = 0.2$ | 73.80 | 93.14 | 73.94 | 93.08 |
| ImageNet-100 | $t = 0.3$ | 72.30 | 92.94 | **74.04** | **93.24** |
| | $t = 0.4$ | 69.92 | 92.10 | 73.12 | 92.98 |
| | $t = 0.5$ | 68.60 | 90.90 | 72.88 | 92.84 |

We further validate the effectiveness of the proposed loss through experiments on the ImageNet dataset. Table 4 reports the top-1 and top-5 classification accuracies on ImageNet-100 for various temperature values. For each temperature, we compare SimCLR with and without the proposed loss. The results demonstrate that incorporating our loss generally improves

*Table 5.* Top-1 accuracy (%) on the full ImageNet dataset.

| Method | Top-1 accuracy (%) |
|---|---|
| SimCLR | 51.79 |
| SimCLR + Ours | 52.48 |

performance across a range of temperatures, with the largest gain observed at $t = 0.3$.

We also report results from 100-epoch training on the full ImageNet dataset. Using the same experimental setup as for ImageNet-100, we compare SimCLR with our method (SimCLR + the proposed loss) using $t = 0.2$ and $\lambda = 40$. As shown in Table 5, our method outperforms the baseline.

## F. Discussion on the Proposed Loss for Variance Reduction

The auxiliary loss $\mathcal{L}_{\text{VRNS}}(\mathbf{U}, \mathbf{V})$ proposed in Def. 5.7 shows effectiveness in the following scenarios:

- **Small Batch Training:** As established in Theorem 5.5, the variance of negative-pair similarities increases as batch size decreases. The proposed loss in Def. 5.7 directly penalizes this variance, making it particularly advantageous when training with small batch sizes. This effect is empirically validated in Figure 4.

- **Temperature Robustness:** The performance of CL methods is often sensitive to the choice of the temperature parameter, which affects the distribution of similarities among embedding pairs (Wang & Liu, 2021). By explicitly encouraging negative-pair similarities towards the optimal value of $-1/(n - 1)$, the proposed loss reduces this sensitivity and stabilizes performance across a wide range of temperature settings, as shown in Figure 3.

Despite its advantages, the proposed loss also presents several limitations:

- **Suppression of Semantically Meaningful Variance:** In some cases, variance in negative-pair similarities may capture meaningful semantic differences between instances. Enforcing uniform similarity can potentially suppress this informative structure, adversely affecting representation quality.

- **Reduced Impact with Large Batch Sizes:** The variance-reducing effect of proposed loss diminishes as batch size increases, since the variance of negative-pair similarities naturally decreases in larger batches.

- **Hyperparameter Sensitivity:** The proposed loss introduces an additional hyperparameter, $\lambda$, which necessitates careful tuning to achieve optimal performance.

