# OpenReview forum: "On the Similarities of Embeddings in Contrastive Learning"
_ICML.cc/2025/Conference — ICML 2025 poster_

### Official Review · Reviewer_Zp1i · 2025-03-10

**Overall Recommendation:** 3

**Summary:**

This paper investigates the geometry of embeddings learned by contrastive learning. This paper first extends the geometry of optimal embeddings (perfectly aligned positives and negatives with cosine similarity $-1/(n-1)$) to an inclusive form of contrastive loss. Then it proves that over-separated negatives with similarity less than $-1/(n-1)$ harm the perfect alignment of positives, and that in fixed mini-batch training, the same-batch negatives are over-separated, especially when the batch size is small. To address this problem, the authors propose a VRN loss term to regularize the similarity of negative pairs. They conduct experiments on benchmark datasets to validate the effect of the proposed VRN.

**Claims And Evidence:**

- In the first contribution (line 55), the authors claim that within-view negative pairs can mitigate the excessive separation of negative pairs under full-batch scenarios, whereas this point is supported by neither specific theorems nor experiments. Moreover, in the single-modal case, the cross-view and within-view pairs seem to have no difference, because they are both generated by the same combination of random augmentations. Do I understand this correctly? If so, how can more negative pairs help reduce the excessive separation?
- To address the excessive separation problem of mini-batch training, the authors derive the theorems under the *fixed* mini-batch assumption. However, many contrastive learning methods (e.g. SimCLR, MoCo, etc.) empirically support the use of *random* mini-batches, i.e., the loader has the data reshuffled at every epoch. Is excessive separation still a problem under the *random* mini-batch scenarios? Or at least the authors should demonstrate the significance of the *fixed* mini-batch setting.
- Proposition 5.2 demonstrates that excessive separation harms perfect alignment, but I think there lack demonstrations of the negative effect of imperfect alignment, because the essential goal is not to minimize the contrastive loss but to achieve good embeddings. What if the excessive separation brings some advantages to the downstream tasks? To demonstrate the negative effect of imperfect alignment, I think perhaps a worse error bound of the downstream generalization (e.g. linear probing) is necessary.
- The authors claim that the proposed VRN loss term improves the accuracy of contrastive losses. However, according to Table 1 and Figure 2, there are cases where the accuracy drops after incorporating VRN, especially in Figure 2 CIFAR-100 DCL, the top 1 accuracy drops in 4 of the 5 cases.

**Essential References Not Discussed:**

The related works are properly cited.

**Experimental Designs Or Analyses:**

Please refer to *Methods And Evaluation Criteria*.

**Methods And Evaluation Criteria:**

- The experiments are conducted based on the InfoNCE-based contrastive losses. Additional validations on the independently additive contrastive loss (Definition 3.2) could be helpful.
- I understand the authors might have limited computational resources, but additional experiments on the full ImageNet can be more convincing.

**Other Comments Or Suggestions:**

The notations need refinement or specification.
1. At the beginning of Section 3, $x$ and $y$ are denoted as two distinct augmentations of the same instance (positive pair). Yet in the following parts, they can also represent negative pairs.
2. The variance of a vector typically means the covariance matrix, but the notation $Var$ in this paper indicates the deviation w.r.t. the l2 norm. This needs specification to improve the readability.

**Other Strengths And Weaknesses:**

**Strengths**
1. This paper extends the theoretical analysis of contrastive losses with proper assumptions on the generation of positive and negative samples.
2. This paper studies from a new perspective and proves that the similarity of negative pairs could affect the alignment of positive pairs.

**Weaknesses**
1. The fixed mini-batch assumption is somewhat less realistic. More evidence is needed to show if the excessive separation of negative pairs is indeed a real problem in contrastive learning.
2. The experimental results are less convincing.

**Questions For Authors:**

1. What is the difference between cross-view and within-view negatives in the single-modal scenario?
2. Is excessive separation still a problem under the *random* mini-batch scenarios?
3. What is the negative effect of imperfect alignment?
4. How does VRN perform under additive contrastive loss and full ImageNet?

**Relation To Broader Scientific Literature:**

NA.

**Theoretical Claims:**

I've checked the proofs of Theorem 5.1, Proposition 5.2, and Theorem 5.6. They seem to be correct.

---

> ### Author Rebuttal · Authors · 2025-04-01
>
> We thank the reviewer for the thoughtful and constructive review, especially the effort to understand and verify our theoretical results. We are glad the reviewer saw our analysis as a meaningful **extension of existing CL theory** and appreciated its **new perspective** on the role of negative pair similarity. Below, we provide detailed responses to your constructive comments.
>
> ---
> ### Q1 & C1-1. The difference between cross-view and within-view negatives in the single-modal scenario?
>
> Thank you for the insightful question. The figure below illustrates the structural difference between cross-view and within-view negatives in the single-modal case. Each graph has six nodes for three instances, with edges denoting negative pairs: $(u_i, v_j)$ for cross-view, and $(u_i, u_j)$ for within-view, where $i\neq j$.
>
> [`Fig.4` Illustration of cross-view and within-view negatives.](https://osf.io/ejbm3?view_only=066d766d57914710810f46ab5f849bf9)
>
> One can confirm that cross-view graphs are **fully connected bipartite**, whereas within-view graphs consist of **disconnected subgraphs**. This topological difference highlights their non-equivalence, even in the unimodal setting.
>
> We will clarify this distinction in the revision.
>
> ---
> ### Q2 & C2. Is excessive separation still a problem under the random mini-batch scenarios?
>
> Yes, as confirmed by additional experiments. In `Table 1` of our response to Reviewer BhMK, excessive separation still emerges under random mini-batching and correlates with performance degradation when it occurs more frequently.
>
> ---
> ### Q3 & C3. The negative effect of imperfect alignment?
>
> Imperfect alignment degrades representation quality. Using the sigmoid loss (in Example 5.4) with varying bias $b$, we observe in the figure below that lower positive similarities correlate with reduced top-1 accuracy in linear probing. This demonstrates the performance sensitivity to alignment quality.
>
> [`Fig.5` Negative effect of imperfect alignment.](https://osf.io/57knr?view_only=066d766d57914710810f46ab5f849bf9)
>
> ---
> ### Q4. VRN performance under additive contrastive loss and full ImageNet?
>
> We evaluated VRN loss in the linear probing setting on CIFAR-10. Using Sigmoid loss alone ($t=1$, $b=-1$) yields 81.10% accuracy, while combining it with VRN loss ($\lambda=30$) improves performance to 87.81%, under the same training setup as in our paper. This demonstrates a significant gain from incorporating VRN, under additive contrastive loss.
>
> We applied VRN to SimCLR on full ImageNet, using the same setup as in our ImageNet-100 experiments. After 100-epoch training with $t=0.2$ and $\lambda=40$, our method (SimCLR+VRN) achieved 52.48% top-1 accuracy, outperforming the SimCLR baseline (51.79%). This confirms the scalability of VRN to large-scale datasets.
>
> ---
> ### C1-2. The contribution, that within-view negative pairs can mitigate the excessive separation of negative pairs, is supported by neither specific theorems nor experiments.
>
> The mitigation effect of within-view negatives is theoretically supported by Theorems 5.1 and 5.3 (see final paragraph of Sec. 5.1). Theorem 5.3, based solely on cross-view negatives $(c_1, c_2) = (1, 0)$, leads to excessive separation due to the independently additive form of the loss. In contrast, the loss in Theorem 5.1 includes both cross-view and within-view negatives $(c_1, c_2) = (1, 1)$, which alleviates this issue.
>
> To empirically validate this, we evaluated the sigmoid loss (as in Example 5.4) with and without within-view negatives. As shown in `Table 2` of our response to Reviewer BhMK, adding within-view negatives reduces the proportion of excessively separated negative pairs, confirming the theoretical insight.
>
> ---
> ### C4. Accuracy drops after incorporating VRN?
>
> The initial drop was due to using a fixed VRN weight ($\lambda=30$) at submission time. After tuning $\lambda$, we observed consistent performance gains across all methods and datasets. Updated results are provided in `Fig.1` of our response to Reviewer BhMK.
>
> ---
> ### S1. Are $(x,y)$ always positive pairs? Later parts seem to allow negative pairs as well.
>
> There was a notational oversight in Sec. 3. The notation $(x,y)$ should represent either positive or negative pairs depending on the sampling. From Sec. 4, we clarify this with $(x,y)\sim p_{pos}$ or $p_{neg}$. We’ll revise Sec. 3 to reflect this properly.
>
> ---
> ### S2. The notation of $Var$.
>
> Thank you for pointing this out. The standard form is $Var[X]=\mathbb{E}[(X-\mathbb{E}[X])(X-\mathbb{E}[X])^\top]$, while we wrote $\mathbb{E}[(X-\mathbb{E}[X])^\top(X-\mathbb{E}[X])]$. We’ll revise the appendix to follow the standard or express it as an expectation.

---

> > ### Comment · Reviewer_Zp1i · 2025-04-05
> >
> > I thank the authors for the detailed reply. The rebuttal solves most of my questions, but I still have major concerns about Q2.
> >
> > By "random minibatch," I mean breaking the fixed minibatch assumption, allowing batches to contain different samples in different epochs. For example, when using torch.utils.data.DataLoader, this could be easily realized by setting "shuffle=True"  to have the data reshuffled at every epoch. (The default setting is "shuffle=False".) In this case, there will be no fixed same-batch and different-batch negative pairs through training, and consequently, it avoids excessive separation.
> >
> > Perhaps I didn't make this point clear enough in my last review. The additional Table 1 in the rebuttal doesn't seem to address the above concerns. Therefore, I suggest the authors conduct experiments with the per-epoch reshuffled dataloader to verify if the excessive separation is indeed a significant problem worth investigating.

---

> > > ### Author Response · Authors · 2025-04-05
> > >
> > > Thank you for the clarification.
> > >
> > > We confirm that **all experiments on real-world datasets used `shuffle=True`**, meaning minibatches were reshuffled at each epoch. This setting was used consistently, **including for `Table 1` and all experiments in Sec.6 of our manuscript**. It can be verified in our code (e.g., [line 396 here](https://anonymous.4open.science/r/vrn/solo/data/pretrain_dataloader.py)).
> > >
> > > We hope this addresses the remaining concern.
> > > Please feel free to add comments if you have any further questions or need additional clarification.

---

### Official Review · Reviewer_qfkv · 2025-03-14

**Overall Recommendation:** 4

**Summary:**

This paper mathematically analyzes the geometric properties of the positive pairs’ embeddings as well as negative pairs’ embeddings in different contrastive learning objectives. The authors mathematically find the optimal threshold for the expected negative pair similarities that results in preventing a misalignment of positive pairs in the embedding space.

They further provide an analysis of how the variance of similarity of the negative pairs can also affect the misalignment of positive samples. This variance is especially evident in the mini-batch setting.

Using the optimal threshold of the similarities of negative samples, the authors propose a comprehensive variance reduction loss function, namely VRN, for the negative pairs that can be added to any existing contrastive loss function. They show its effectiveness in classification tasks using various contrastive learning loss functions on the CIFAR and ImageNet datasets.

**Claims And Evidence:**

Yes, the claims seem convincing to me.

**Essential References Not Discussed:**

None I am aware of.

**Experimental Designs Or Analyses:**

Yes. The paper provides experiments on 3 classification datasets and examines them with regard to the classification accuracy.

However, the paper could benefit from more in-depth analysis of embeddings and the structure of the embedding space after the proposed VRN approach. E.g., analysis of the distribution of cosine similarities of the positive and negative samples and PCA/DOSNES visualizations based on classes could be provided. These analysis would provide more insights on the geometric structure of the embedding space.

**Update after author response:** The additional experiment analysing the embedding space seems helpful, but additional empirical analysis of typical real-world models, which are based on much larger training runs, would be a welcome addition, especially also with further insights regarding the effects on the semantics in the vector space.

**Methods And Evaluation Criteria:**

Yes, the evaluation makes sense. The authors also provide a link to the source code.

**Other Comments Or Suggestions:**

The notation d in lines 242, 283 and 313 seem to not have been introduced in the paper.

**Other Strengths And Weaknesses:**

N/A

**Questions For Authors:**

None

**Relation To Broader Scientific Literature:**

The findings appear to provide an important insight for contrastive learning in unimodal and multimodal settings. One of the pioneering works in this direction is [1], and since then, many efforts have been done to understand the alignment and separation of embeddings in the embedding space from a geometrical point of view and their results on downstream tasks.

[1] Wang, Tongzhou, and Phillip Isola. "Understanding contrastive representation learning through alignment and uniformity on the hypersphere." International conference on machine learning. PMLR, 2020.

**Theoretical Claims:**

I did not attempt to check the proofs.

---

> ### Author Rebuttal · Authors · 2025-04-01
>
> We thank the reviewer for acknowledging that our mathematical analysis **provides important insights for contrastive learning in both unimodal and multimodal settings**. We are also encouraged by the positive evaluation of our experimental claims and the effectiveness of our proposed VRN approach on classification tasks. Following the reviewer’s constructive suggestions to strengthen our paper, we provide additional experiments as follows.
>
> ---
> ### E1. More in-depth analysis of embeddings and the structure of the embedding space after the proposed VRN approach.
>
> To analyze the effect of the proposed VRN loss on the embedding space, we conduct experiments on both synthetic and real-world datasets.
>
> **[Experiment on Synthetic Data]**
>
> We use a synthetic dataset of 4 samples, each augmented twice (8 embeddings in 3D). Embeddings are optimized directly using SGD (lr=0.5, 100 steps, mini-batch size=2). We compare two setups: (1) SimCLR with temperature $t=0.5$, and (2) SimCLR + VRN loss with $\lambda=3$.
>
> [`Fig.3`. Visualization of learned embeddings.](https://osf.io/phte3?view_only=066d766d57914710810f46ab5f849bf9)
>
> As shown in the figure above, both methods successfully align positive pairs. However, with SimCLR+VRN, the negative pairs are more evenly distributed in cosine similarity, centering around the theoretical optimum of $-1/3$. Specifically:
>
> - SimCLR: mean = -0.3201, std = 0.2051
> - SimCLR+VRN: mean = -0.3327, std = 0.0207
>
> Given the absence of semantic structure in the synthetic samples, a uniform separation among negative embeddings is desirable. The VRN loss seems to facilitate such balanced distribution.
>
> **[Experiment on Real Data]**
>
> We further assess how VRN affects negative pair distribution in realistic scenarios. Using ResNet encoders trained on CIFAR100 with either SimCLR or SimCLR+VRN, we sample 5,000 negative pairs from augmented training images and compute their cosine similarities.
>
> We repeat this across different batch sizes and report the variance of negative-pair similarities:
>
> | Batch size | Variance of the negative-pair similarity (SimCLR) | Variance of the negative-pair similarity (SimCLR+Ours) |
> |:-:|:-:|:-:|
> |32 |0.1649|0.1008|
> |64 |0.1505|0.0952|
> |128|0.1444|0.0929|
> |256|0.1404|0.0921|
> |512|0.1396|0.0917|
>
> `Table 3` *The effect of VRN loss on the variance of negative-pair similarity, where embeddings are generated from models pretrained with different batch sizes.*
>
> We observe that the variance of negative-pair similarity consistently decreases when VRN is used, across all batch sizes. We will include these experimental results and their discussion in the revised version of the manuscript.
>
> ---
> ### S1. The notation d in lines 242, 283 and 313 seem to not have been introduced in the paper.
>
> Thank you for pointing this out. The notation $d$ refers to the dimension of embedding vectors, as mentioned in line 96 in Sec. 3 (Problem Setup). We will clarify this in the relevant lines.

---

### Official Review · Reviewer_BhMK · 2025-03-16

**Overall Recommendation:** 3

**Summary:**

The paper analyzes the distribution of positive and negative pairs in contrastive learning and shows that perfect alignment becomes impossible when expecting negative pairs to fall below the optimal threshold. The paper also proposes variance reduction for negative-pair similarity loss to reduce the variance of negative pairs when using a small batch size. Furthermore, they experiment on different datasets such as CIFAR-100 and ImageNet-100 to demonstrate that this addition can improve contrastive learning methods.

**Claims And Evidence:**

The papers claims that

**Essential References Not Discussed:**

The papers mentiones related works adequantly.

**Experimental Designs Or Analyses:**

All the experimental section. I am wondering how many negative pairs falles above the optimal threshold in the experimnts.

**Methods And Evaluation Criteria:**

The paper provides comparison on CIFAR-10, CIFAR-100 and ImageNet-100 datasets using 4 methods. The numerical results show that using VRN increase the accuraccy in most cases.

**Other Comments Or Suggestions:**

No other comments.

**Other Strengths And Weaknesses:**

**Strengths**:

1.  The paper is well-written and the results are clearly presented.

**Weaknesses**:
1. The authors did not discussed in which scenarios adding VRN loss could downgrade the performance. For example in Figure 2, I observe that combining VRN with DCL results decrease (Downwards arrows) in most cases.

2. I expect more investigation on the limitations on using VRN and scenarios where it is benefitial to use it.

**Questions For Authors:**

I have the following quesitons from the authors:

1. Could authors provide that how many of negative pairs does not satisfy the optimal thershold if a method only uses negative pairs into the loss function?

2. Is there any toy experiments that we can observe the effect of using VRN loss? (Probably something similar to Figure 1 in [1])


[1] Liu et al., Generalizing and Decoupling Neural Collapse via Hyperspherical Uniformity Gap, ICLR 2023

**Relation To Broader Scientific Literature:**

The paper showed that pervect alignmet becomes unreachable if we expect all the negative pairs to fall bellow the optimal threshold. Instead, considering the negative pairs in the loss function can be more benefitioal. These findings are useful in the future direction research in Contrastive Learning.

**Theoretical Claims:**

I checked proofs for Proposition B.1 and B.2. and they were correct.

---

> ### Author Rebuttal · Authors · 2025-04-01
>
> We appreciate the reviewer’s positive feedback, including that our paper is **well-written**, provides **a clear explanation of results**, and may be **useful for future CL research**. Below, we address each of the reviewer’s comments in detail.
>
> ---
> ### W1. When VRN loss term downgrades the performance in Fig.2?
>
> Fig. 2 in our paper has been updated (see below) with improved hyperparameter tuning:
>
> [`Fig.1` Effectiveness of VRN.](https://osf.io/mejs6?view_only=066d766d57914710810f46ab5f849bf9)
>
> In earlier results, VRN degraded performance on CIFAR-100 (e.g., DCL, DHEL), likely due to a fixed weight $\lambda=30$. After tuning over a wider range of {0.1, 0.3, 1, 3, 10, 30, 100}, we observe consistent improvements across all methods.
>
> We also note:
> * VRN operates on cross-view negatives,
> * SimCLR includes such pairs, DHEL does not,
> * and gains from VRN are larger when the base loss includes cross-view negatives (e.g., SimCLR).
>
> This suggests VRN is most effective when complementing losses that already involve cross-view negatives.
>
> ---
> ### W2. Limitations on using VRN and scenarios where it is benefitial to use it.
>
> We summarize below the scenarios where VRN is most effective, followed by its main limitations.
>
> 1. When VRN is beneficial
> * Small batch training:
> From Theorem 5.5, small batches lead to higher variance in negative-pair similarities. Since VRN minimizes this variance, it yields stronger gains in low-batch regimes. Empirical results in `Fig.1` support this.
> * Robustness to temperature:
> Contrastive loss is sensitive to the temperature parameter, which affects embedding similarity distributions `[R1]`. VRN encourages negative similarities toward the optimal $-1/(n-1)$, reducing performance fluctuation.
> Below, we show SimCLR with VRN ($\lambda=30$) yields more stable accuracy across temperature values on CIFAR-10/100:
>
> [`Fig.2` Robustness to temperature.](https://osf.io/cp3yn?view_only=066d766d57914710810f46ab5f849bf9)
>
> `[R1]` *Wang, et al. Understanding the behaviour of contrastive loss. CVPR2021.*
>
> 2. Limitations of VRN
> * Loss of meaningful structure:
> Variance in negative similarities can reflect semantic diversity. Forcing uniform similarity may suppress this, as discussed in Sec. 4 (L233–245) and Sec. 7 (L433–436).
> * Diminished effect with large batches:
> As batch size increases, the natural variance stabilizes, reducing the marginal benefit of VRN.
> * Hyperparameter tuning:
> VRN introduces an additional weight $\lambda$, which requires tuning per setting.
>
> ---
> ### E1. How many negative pairs fall above the optimal threshold?
>
> We evaluated the ratio of negative pairs whose cosine similarity falls below the theoretical threshold of $-1/(n-1)$, using models pretrained with SimCLR (ResNet-18, CIFAR-100, $t=0.2$).
>
> We generated 5,000 negative pairs from each model by applying random augmentations and measuring cosine similarity from the projector output. The table below summarizes the results:
>
> |Batch size|Ratio below threshold (%)| Var. of negative similarity | Top-1 acc. (%)|
> |:-:|:-:|:-:|:-:|
> |32|58.09|0.1649|56.34|
> |64|58.10|0.1505|58.40|
> |128|57.79|0.1444|58.80|
> |256|57.61|0.1404|59.72|
> |512|57.38|0.1396|59.69|
>
> `Table 1` *Ratio of excessively separated negative pairs and associated statistics.*
>
> We find that over 57% of negative pairs fall below the optimal similarity across all batch sizes. Smaller batches show greater variance and slightly lower accuracy, aligning with Theorem 5.5, which predicts increased variance in negative similarity with reduced batch size.
>
> ---
> ### Q1. How many of negative pairs does not satisfy the optimal thershold if a method only uses negative pairs into the loss function?
>
> We interpret your question in two possible ways:
> (1) How many negative pairs fall below the optimal similarity threshold when training only with the VRN loss, or
> (2) when using a contrastive loss with only cross-view negatives.
>
> Regarding (1), the VRN loss is not intended to serve as a standalone training objective. It is designed to regularize standard contrastive losses, and cannot learn meaningful representations on its own.
>
> Regarding (2), we report statistics from models trained with sigmoid loss (in Example 5.4), which uses only cross-view negatives. In this case, a large fraction of negative pairs have similarities below the threshold of $-\frac{1}{n-1}$, where $n$ is the sample size. This indicates over-separation. When the loss is modified to include both cross-view and within-view negatives, this issue is mitigated, as shown below:
>
>
> |Negative-pair type|Ratio below threshold (%)|
> |:-:|:-:|
> |cross-view|69.30|
> |cross-view & within-view|66.88|
>
> `Table 2` *Negative pair similarity in models pretrained with sigmoid loss.*
>
> Please let us know if this interpretation differs from your intent — we would be glad to clarify further.
>
> ---
> ### Q2. Is there any toy experiments?
>
> Please see `E1` in our response to Reviewer qfkv.

---

### Decision · Program_Chairs · 2025-05-01

**Decision:**

Accept (poster)

**Comment:**

This paper theoretically studies contrastive learning when negative pair are too close to each other, and mitigates the issue of the suboptimality by the variance reduction. Though a few reviewers initially had a concern about how the theoretical/empirical findings will be affected when it comes to the mini-batch setting, the authors have adequately addressed their concerns. Thus, I recommend this submission for acceptance.